# Manipulation Concept: Towards Deriving Generalizable and Physics-informed Manipulation Knowledge of Articulated Objects

## Abstract

Gripper-based articulated object manipulation requires robots to reason about both object structure and the physical constraints of grippers, yet prior work has paid little consideration to grippers' unique characteristics and their interaction with object structures. To alleviate this gap, we introduce **Manipulation Concept**, a novel analytic representation that encodes gripper manipulation skills as parameterized program templates. Each concept formalizes the interaction between a specific actionable part featuring structure and semantic (*e.g.*, cuboid door, ring handle) and a gripper action (*e.g.*, push, lift), linking geometries and semantics with executable robot actions. Building on this representation, we develop an end-to-end framework that (i) leverages a vision-language model to select the most suitable concept for the actionable part, (ii) estimates geometric and affordance parameters to instantiate the selected concept and ground it in the physical world, and (iii) generates precise gripper-specific actions to complete the task. Extensive experiments in simulation and real-world demonstrate that our method outperforms prior approaches in both accuracy and generalization, achieving stronger generalization across object categories, and reliable execution in manipulation tasks.

## 1 Introduction

Articulated objects, composed of rigid and semantically meaningful components connected by joints with translational and rotational motion, are often interacted by humans and embodied agents. Articulated object manipulation is a fundamental task in embodied AI, requiring a broad range of object-interaction skills from grasping and tool use to complex environmental engagement. Among robotic end-effectors, parallel grippers are a common choice due to their versatility and effectiveness in object manipulation (Pagoli et al., 2021), and manipulation using grippers has been extensively studied (Mo et al., 2021; Ning et al., 2023; Huang et al., a; Geng et al., 2023b; Geng et al.; 2023a; Fang et al., 2023; Tang et al., 2023; 2025; Wu et al.).

Despite the extensive use of grippers in object manipulation research, prior work has put emphasis on affordance perception (Mo et al., 2021; Ning et al., 2023; Geng et al., 2023a;b) and object-level task execution (Bu et al., 2025; Kim et al., 2025; Zhen et al., 2024), while paying less attention to how grippers themselves perform manipulation skills and interact with objects. Gripper actions are typically either (i) derived from action representations that generalize across end-effectors (Mo et al., 2021; Ning et al., 2023; Bu et al., 2025; Kim et al., 2025), or (ii) computed based on the structured representation of actionable parts (*e.g.*, axes, bounding boxes) (Geng et al., 2023b; Huang et al., a). However, these approaches either fail to account for the unique characteristics of grippers (Mo et al., 2021; Ning et al., 2023; Tang et al., 2023; Wu et al.) or to find it difficult to provide the fine-grained affordance needed to guide precise actions (Huang et al., a; Geng et al., 2023b;a), leading to unsuccessful manipulation.

To this end, we propose **Manipulation Concept**, a novel analytic representation that encapsulates gripper-based manipulation skills into program-formed and parameterized templates, with its design illustrated in Fig. 1. Each skill refers to the interaction strategy between a specific actionable part and a gripper. The actionable part captures both the geometry and semantics (*e.g.*, `Cuboid Door`, `Ring Handle`, *etc.*), while the interaction strategy specifies how the gripper interacts with

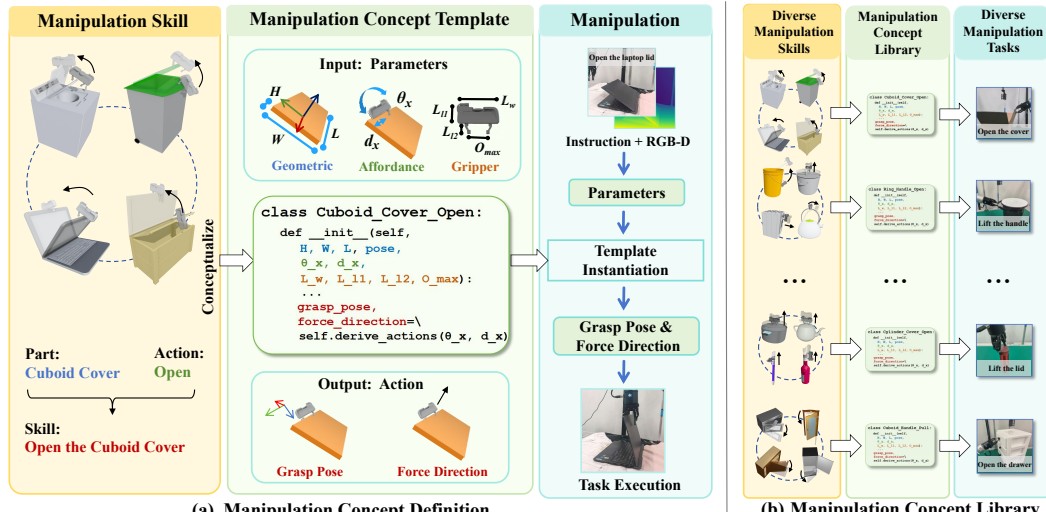

Figure 1: We propose **Manipulation Concept**, an analytic representation that encapsulates versatile manipulation skills by leveraging the intrinsic commonalities of a parallel gripper's interactions. Each concept is a parameterized program template that can be instantiated using geometric and affordance parameters estimated from point clouds. This process physically grounds the manipulation knowledge within the manipulation concept to specific tasks, enabling a robot to perform precise actions with strong generalization across objects and scenarios. A Manipulation Concept Library (MCL) is constructed to support a wide range of tasks.

it (*e.g.*, `Push`, `Lift`, *etc.*). By jointly considering the object's geometric structure and the gripper's physical characteristics, manipulation concept explicitly encodes knowledge for deriving physics-informed actions for gripper-based manipulation tasks, providing a unified representation that links objects' structural properties with appropriate interactions. To support broad applicability across diverse articulated object manipulation tasks, we construct a Manipulation Concept Library (MCL) comprising 52 Manipulation Concepts that cover 18 common types of actionable parts.

Manipulation concept exhibits several appealing properties: **(1) Generalizability.** Its parameterized design enables generalization across objects with diverse geometries that share common spatial properties, allowing the inference of suitable actions for various manipulation tasks. **(2) Scalability.** The programmatic formulation of manipulation concepts facilitates their construction across diverse object geometries and enables alignment between manipulation skills and shape-specific characteristics. **(3) Accuracy.** Manipulation concepts integrate structural properties of actionable parts with their associated affordances to infer precise action proposals for gripper-based manipulation.

To leverage manipulation concepts for proposing gripper actions in manipulation tasks, we introduce an end-to-end framework that integrates these concepts with parameter estimation modules to instantiate them and infer gripper-specific actions for completing manipulation tasks. The framework begins with a VLM that selects the most suitable concept from the manipulation concept library, based on the target actionable part identified from the input object image and the natural language task description. The target part is then grounded using Grounded-SAM (Ren et al., 2024) and mapped to 3D point clouds, which are processed by the parameter estimation module to predict both geometric and affordance parameters for instantiating the chosen manipulation concept. Finally, the instantiated template infers a gripper-specific action that guides the robot in executing the manipulation task.

We conduct extensive experiments to evaluate our proposed framework. In simulation on the large-scale articulated object dataset PartNet-Mobility (Xiang et al., 2020), our method shows its superiority over both classic gripper-based methods (Mo et al., 2021) and state-of-the-art approaches (Huang et al., a) in successfully completing articulated object manipulation tasks. Additionally, we conduct real-world experiments and the high success rates demonstrate the practicality and generalizability of our approach.

In summary, our contributions are as follows: **(1)** We propose **manipulation concept**, a novel analytic representation that encodes manipulation skills into parameterized program templates, unifying affordances across diverse articulated object manipulation tasks into adaptive robotic actions. **(2)** We develop an end-to-end framework that leverages manipulation concepts to infer gripper-specific action proposals for manipulation tasks, enabling robots to robustly perform object manipulations under natural language instructions. **(3)** To instantiate manipulation concepts, we design parameter estimation modules that infers both geometric and affordance parameters, grounding the structural and functional properties of objects for use within manipulation concepts.

## 2 RELATED WORK

### 2.1 ARTICULATED OBJECT MANIPULATION

Articulated object manipulation remains challenging due to the inherent diversity in geometric shapes, semantic categories, and kinematic structures (Cui et al., 2025). Existing approaches mainly fall into two categories: affordance-based methods and articulation estimation (Kim et al., 2024).

Affordance-based methods (Mo et al., 2021; Ning et al., 2023; Wu et al.) generate dense 2D affordance heatmaps to identify probability in manipulation pixels. While these heatmaps guide contact point selection and action prediction, they often rely on coarse 2D representations and binary movable-part masks (Mo et al., 2021), limiting the precision in part localization and interaction. Other approaches utilize articulation estimation to facilitate generalizable manipulation. These methods employ various representations, such as the 6D pose of actionable parts (Geng et al., 2023b), joint parameters (*e.g.*, joint type and axis) (Huang et al., a; Yu et al., 2024; Wang et al., 2024b), or articulation flow (Eisner & Zhang, 2022; Zhang et al.) to capture the object's articulation properties. The obtained representations can then be combined with heuristic methods (Geng et al., 2023b) to perform manipulation. However, they are often insufficient on their own to provide the fine-grained affordance information necessary to guide precise, task-specific actions or to maintain adaptability when faced with geometric variations. By introducing the manipulation concept, our framework can first accurately locate the movable part and then precisely specify the gripper's interaction with the part for downstream manipulation task.

### 2.2 STRUCTURED REPRESENTATIONS FOR MANIPULATION

Structured representations are critical for enabling effective and robust manipulation. Various representation types have been explored for robot manipulation, including: *Keypoint* (Huang et al., b; Manuelli et al., 2019; Zhong et al., 2023; Liu et al., 2025), *6D pose* (Geng et al., 2023b; Pan et al., 2025; Geng et al.; 2023a; Liu et al., 2023), *Joint parameter* (Huang et al., a; Yu et al., 2024; Wang et al., 2024a), *Flow* (Eisner & Zhang, 2022; Zhang et al.; Yuan et al., 2025; Xu et al.). Structured representations offer several benefits, including (1) facilitating generalizable manipulation by enabling knowledge transfer across instances sharing similar representations (Geng et al., 2023b; Cui et al., 2025), (2) expanding the range of usable training data (Huang et al., a; Wen et al., 2023), (3) providing physical meaning and thus allowing integration of prior knowledge and compatibility with LLM or VLM for reasoning (Sun et al., 2025; Huang et al., c). These methods typically involve a two-stage pipeline (Yang et al., 2025): a perception module extracts structured representations, followed by a policy that maps these to actions. In contrast, our proposed manipulation concept encodes the manipulation knowledge about a skill. By instantiating these pre-defined program templates with estimated parameters (Zhang et al., 2025; Wei et al., 2024), the concepts directly derive precise gripper actions directly from the selected concept.

## 3 METHOD

### 3.1 MANIPULATION CONCEPT

A manipulation concept is a parameterized program template that encodes manipulation skill knowledge about how a gripper interacts with specific actionable parts. For example, as shown in Fig. 2, a parallel gripper is queried to lift a ring-shaped handle with an upward force direction, while the grasp pose can vary along the ring, with the variation controlled by parameters. We formalize such

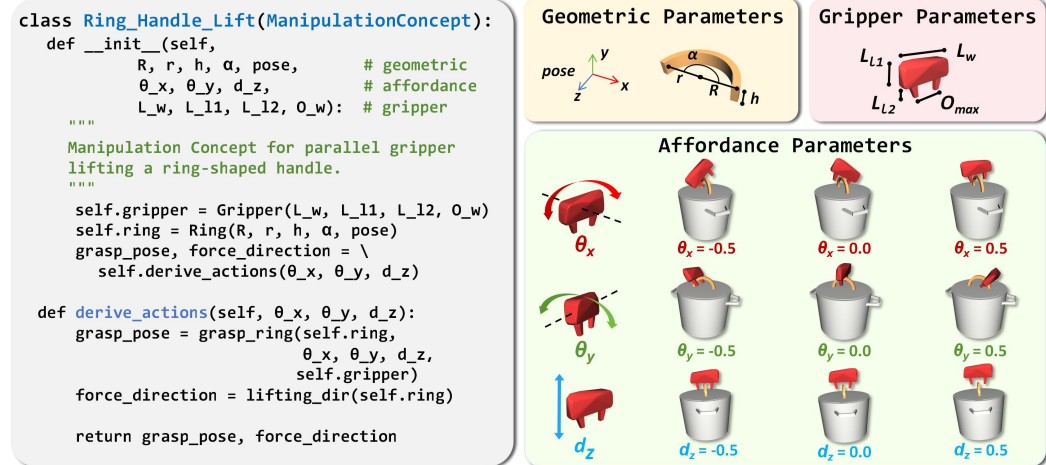

Figure 2: Illustration of the **Ring_Handle_Lift** manipulation concept. **Left:** its definition in the form of a code template. **Right:** the visualization of its parameters' effect. The geometric parameters control the size and pose of ring primitive, while the gripper parameters represent the gripper's fixed dimensional properties. The affordance parameters $\theta_x$, $\theta_y$, $d_y$ specify the gripper's precise rotational adjustments and translational offset relative to the ring. Only the grasp pose is visualized; the force direction is omitted as it is always upward.

prior knowledge into a series of spatial transformations, creating a parameterized program that takes input parameters describing the geometric structure of the actionable part and the gripper-part interaction pattern, and then outputs a grasp pose and a force direction to guide the manipulation. As shown in Fig. 2, the parameters of manipulation concept fall into three categories: **geometric parameters** specifying the structure of **what** to manipulate, **affordance parameters** defining **how** to manipulate, and **gripper parameters** defining **with what** to manipulate. Due to the parameterized nature of manipulation concepts, they can capture variations within single manipulation skills, and enable the incorporation of learning mechanisms for deriving flexible interaction strategy.

**Geometric Parameters.** These parameters describe the geometric structure of the actionable part using a set of common parameterized 3D geometric primitives (*e.g.*, cuboid, sphere, ring, cylinder), as we have found that these simple shapes are sufficient for most manipulation tasks. This approach allows us to abstract away complex geometric details and provide the essential structural information for manipulation. For example, the geometric parameters for a **Cuboid_Door_Open** concept include the door's length, width, height, and a 6-DoF pose.

**Affordance Parameters.** Unlike coarse 2D affordance maps that only indicate *where* to interact, our affordance parameters directly specify *how* an action should be performed on a geometric structure. Given the estimated geometric parameters, manipulation concept leverages its encoded prior manipulation knowledge to define a feasible interaction space. Then the affordance parameters indicate how to effectively interact within this feasible space. The template contains multiple affordance parameters, each normalized to the range $(-1, 1)$. These parameters act as proportional values that control the gripper's grasp pose. A specific combination of these parameters uniquely determines a precise manipulation mode (*i.e.*, a specific grasp pose and force direction). By adjusting these parameters, a single template can encompass most feasible interaction modes, thereby endowing the manipulation concept with high flexibility and adaptability.

**Gripper Parameters**. To analytically compute the feasible end-effector pose for the gripper, we utilize gripper parameters to encapsulate essential structural information of the parallel gripper, such as the length of the fingers and the maximum opening width. This explicitly allows the manipulation concept to incorporate the physical constraints of the specific gripper.

This parameterized design allows manipulation concept to be adaptively instantiated based on various perceptual information, thereby enabling the physical grounding of general manipulation knowledge into concrete robot actions, rather than relying on static, heuristic rules.

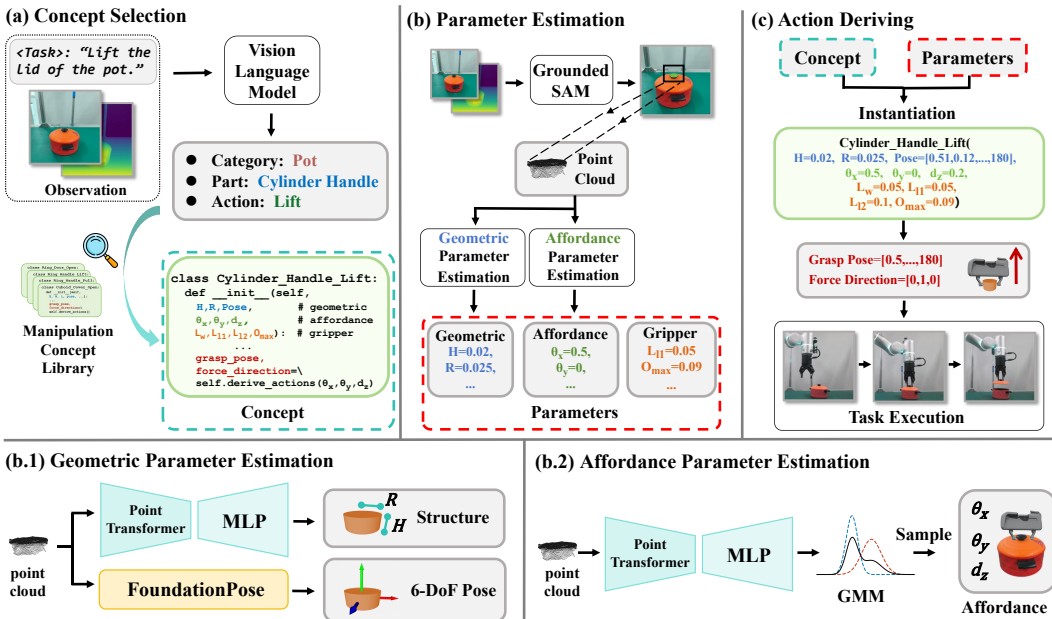

Figure 3: Overview of our framework. Given an RGB-D image and language instructions, our framework first leverages a Vision-Language Model (VLM) to extract key task information, including the object category, the part to be manipulated, and the required action. Based on this information, we select the most suitable manipulation concept from our pre-built library. We then use Grounded-SAM (Ren et al., 2024) to obtain the point cloud of the target part, which is fed into our proposed parameter estimation module for concept instantiation. This process allows us to derive the precise actions required for the task.

In this way, we have systematically designed and constructed a Manipulation Concept Library (MCL) containing 52 distinct concepts across different combinations of 18 common types parts with various actions, covering a broad range of articulated object manipulation skills. When faced with an unseen task, we can retrieve these concepts from the library to accomplish the manipulation in a flexible and efficient manner.

## 3.2 FROM MANIPULATION CONCEPT TO EXECUTABLE ACTION

Building on manipulation concepts, we propose a complete framework for articulated object manipulation, as illustrated in Fig. 3. The framework takes as input task description in natural language and an RGB-D image containing the target object. In Sec. 3.2.1, we describe how a vision-language model (VLM) selects the most suitable manipulation concept for the target actionable part from the manipulation concept library. In Sec. 3.2.2, we detail the detection and segmentation of the actionable part. In Sec. 3.2.3, we describe how the parameters for instantiating the selected manipulation concept are acquired, thereby enabling the grounding of these concepts in the physical world. Finally, in Sec. 3.2.4, we show how the instantiated manipulation concept guides executable robot action.

### 3.2.1 CONCEPT SELECTION

Given a manipulation task description in natural language, our method first selects the most suitable manipulation concept that describes the actionable part from our pre-built manipulation concept library. The selection is guided by three pieces of information extracted from the task description and the scene's RGB image using a VLM. **1) Object Category** identifies the general type of the target object (*e.g.*, "Bottle", "Laptop", *etc.*), and provides initial context and narrowing possible concept candidates. **2) Part Type** specifies the actionable part of the target object for completing the task (*e.g.*, "Cylinder Cap", "Cuboid Cover", *etc.*), capturing both geometric and semantic properties.

**3) Action Type** defines the required action for completing the manipulation task on the actionable part (*e.g.*, "pull", "lift", *etc.*)

Together, these three factors uniquely map to one manipulation concept in the MCL. For example, combining "cylinder cap" with "lift" corresponds directly to the "Cylinder_Cap_Lift" concept, which is then selected for execution.

### 3.2.2 ACTIONABLE PART GROUNDING

After the task-relevant information is acquired in Sec. 3.2.1, the part type is used to construct a text query for Grounded-SAM (Ren et al., 2024) to extract a segmentation mask for the target part. The resulting mask is used to crop the depth image, which is then reprojected to a single-view partial point cloud using camera intrinsics. This part-level point cloud serves as the direct input for the subsequent parameter estimation module.

### 3.2.3 PARAMETER ESTIMATION FOR CONCEPT

To instantiate the selected concept, we design a parameter estimation module to estimate geometric parameters and affordance parameters. Gripper parameters are provided a priori, as they are directly available from the robot's URDF file.

**Geometric Parameter Estimation.** Geometric parameters are used to describe a 3D primitive's physical structure, consisting of two components: structural parameters and 6-DoF pose parameters.

**1. Structural parameters.** We first employ a Point Transformer (Zhao et al., 2021) encoder that takes a part-level point cloud as input. The extracted feature is then passed to an MLP to predict the structural parameters of the corresponding 3D geometric primitive (*e.g.*, the length, width, and height of a cuboid). Each geometric primitive type is associated with a dedicated MLP, since the parameters for each primitive have different lengths and meanings.

**2. 6-DoF pose parameters.** We leverage the FoundationPose (Wen et al., 2024) to estimate the 6-DoF pose of the target part in the global coordinate system. In this model-based setting, FoundationPose (Wen et al., 2024) requires a 3D CAD model of the object for reference. To provide this, we first generate a canonical mesh at a standard pose using the estimated geometric parameters. This mesh, together with RGB-D image and the mask of the target part, is then fed into FoundationPose (Wen et al., 2024) for 6-DoF pose estimation.

Finally, by combining the predicted 6-DoF pose parameters with the 3D geometric primitive's structural parameters, we obtain the whole geometric parameters of the target part. Please refer to Appendix A.1.4 for further details.

**Affordance Parameter Estimation.** A specific gripper-part interaction may involve multiple valid modes, leading to a multimodal distribution for affordance parameters. We address this multimodality by representing the affordance parameter space as a Gaussian Mixture Model (GMM), allowing us to sample from a distribution of feasible solutions rather than regressing to a single value. We adopt a network architecture similar to that used for structural parameters estimation: a Point Transformer (Zhao et al., 2021) encodes the part-level point cloud, and outputs the parameters of a GMM via an MLP. Each concept is associated with a dedicated MLP. Please refer to Appendix A.1.5 for further details.

### 3.2.4 CONCEPT INSTANTIATION FOR ACTION DERIVING

With the manipulation concept selected and its parameters estimated, we proceed to instantiate the concept, thereby achieving physically-grounded manipulation. By analytically computing the spatial and geometric transformations necessary for precise gripper-part interactions, the programmatic template derives the parameters for executable robot actions, including the grasp pose and force direction. The robot then executes these actions: the end-effector, while maintaining the specified gripper width, first moves to the grasp pose, executes the grasp, and then moves a predefined distance along the force direction.

### 3.3 IMPLEMENTATION DETAILS

#### 3.3.1 DATA COLLECTION

To prepare training data for both geometric and affordance parameters estimation, we leverage the annotations provided by ConceptFactory (Sun et al., 2024), which labels each articulated object in the PartNet-Mobility dataset (Xiang et al., 2020) with concept identity and corresponding parameter information of its proposed Analytic Concept. To align with the geometric parameters of our manipulation concepts, which describe 3D geometric primitives to represent target parts, we programmatically convert the rich but overly complex original concept-level parameters into a simpler format. This crucial translation step abstracts the key geometric information necessary for manipulation while discarding irrelevant details. For example, a washing machine door, which contains multiple complex shapes in annotation, is simplified into a parameterized ring primitive.

**Geometric Parameters.** We begin by importing PartNet-Mobility objects into the SAPIEN simulator (Xiang et al., 2020) and acquiring their single-view partial point cloud data using a camera in the scene. We leverage SAPIEN (Xiang et al., 2020)'s part segmentation API to precisely mask and extract the point cloud belonging solely to the target manipulable part. The target part's point clouds serve as the network's input, with the corresponding simplified geometric parameters from the annotation translation step serving as our supervisory labels.

**Affordance Parameters.** We adopt an offline interaction strategy (Mo et al., 2021) due to the difficulty of manual annotation. We uniformly sample affordance parameters within their defined range. These sampled parameters, combined with the ground truth geometric parameters (translated from annotations) are then passed to the selected concept to generate diverse executable actions. Only successful interactions are retained as positive samples, where success is judged by the metric detailed in Sec. 4.1.1. For each successful interaction, we record the target part's partial point cloud as the network's input, with the associated affordance parameters as the supervisory labels for model training.

Please refer to Appendix A.1.6 for further details on data collection.

## 4 EXPERIMENTS

To evaluate the effectiveness and generalizability of our method, we conduct extensive experiments in simulation, where SAPIEN (Xiang et al., 2020) is adopted as the simulator. We select 24 common object categories from PartNet-Mobility (Xiang et al., 2020), and further divide them into 16 categories for training and 8 for testing. To further validate our method's robustness, we perform a series of real-world experiments on common household objects.

Table 1: Comparisons of our method against baseline methods on PartNet-Mobility dataset. All experiments are conducted under the parallel gripper setting.

| Method | Train Categories | | | | | | | | | | | | |
|---|---|---|---|---|---|---|---|---|---|---|---|---|---|
| Where2Act | 0.13 | 0.15 | 0.28 | 0.31 | 0.18 | 0.31 | 0.11 | 0.40 | 0.32 | 0.08 | 0.26 | 0.20 | 0.38 |
| ManipLLM | 0.24 | 0.28 | 0.39 | 0.38 | **0.55** | 0.45 | 0.45 | 0.77 | 0.39 | 0.15 | 0.39 | **0.46** | 0.58 |
| A3VLM | 0.25 | 0.35 | 0.54 | 0.26 | 0.49 | 0.58 | **0.59** | 0.49 | **0.75** | 0.21 | 0.50 | 0.42 | 0.29 |
| Ours | **0.78** | **0.43** | **0.80** | **0.53** | 0.51 | **0.59** | 0.39 | **0.82** | 0.53 | **0.34** | **0.79** | 0.25 | **0.58** |

| Method | Train Categories | | | AVG | Test Categories | | | | | | | | AVG |
|---|---|---|---|---|---|---|---|---|---|---|---|---|---|
| Where2Act | 0.18 | 0.16 | 0.30 | 0.24 | 0.13 | 0.19 | 0.07 | 0.29 | 0.31 | 0.26 | 0.24 | 0.15 | 0.20 |
| ManipLLM | 0.21 | 0.21 | 0.49 | 0.40 | 0.26 | 0.37 | 0.33 | 0.38 | 0.42 | 0.43 | 0.43 | 0.37 | 0.37 |
| A3VLM | 0.46 | 0.67 | 0.51 | 0.46 | 0.38 | **0.66** | 0.23 | 0.37 | 0.46 | 0.41 | 0.39 | 0.40 | 0.41 |
| Ours | **0.75** | **0.67** | **0.76** | **0.59** | **0.57** | 0.52 | **0.49** | 0.45 | **0.67** | 0.55 | **0.79** | **0.64** | **0.58** |

### 4.1 SIMULATION EVALUATION

#### 4.1.1 EXPERIMENT SETTING

Our experimental setup follows Where2Act (Mo et al., 2021). In our experiments, an articulated object is positioned with its base fixed and its joint state initialized randomly. Given a randomly sampled RGB-D observation, the agent's task is to interact with a designated movable part to induce a specified motion. To facilitate a more rigorous and fair evaluation, certain configuration for baseline methods (Li et al., 2024; Huang et al., a) are adapted. Detailed experiment settings are provided in the Appendix A.2.1.

We adopt the success rate as our evaluation metric. A successful interaction is considered successful if the movable part of the object moves a distance exceeding a set threshold in the same direction as the gripper's motion. Following A3VLM (Huang et al., a), we attempted operations in two opposite directions separately for each test. The task is recorded as a success if it succeeds in either try.

#### 4.1.2 EVALUATION RESULTS

We compare our method against three representative baselines under the same experimental settings. All three methods use parallel gripper as the end-effector, which is compared with ours in Table 1.

(1) Where2Act (Mo et al., 2021): Given a point cloud input, the method computes a per-point score and selects the contact point as the one with the highest score. Additionally, it predicts 100 end-effector orientations, selecting the orientation with the highest score to acquire the contact pose.

(2) ManipLLM (Li et al., 2024): Given an RGB image and a language task description, It uses a vision-language model to predict the contact point and forward direction of a suction gripper. To ensure a fair comparison, we replaced the suction grippers with parallel grippers.

(3) A3VLM (Huang et al., a): Given an RGB image and a language task description, it locates a part of the object and predicts articulation information, which can be translated into robot actions using simple action primitives. To the best of our knowledge, it is the current state-of-the-art method with publicly available code on this benchmark. Since the A3VLM's code about manipulation is not publicly available at the time of this paper's submission, we replicate its method with implementation details in Appendix A.2.2.

As shown in Table 1, our method achieves a significant performance improvement over all three baselines, especially on unseen categories. Specifically, we observe a relative improvement of 28% and 41% over the best baseline on the training and testing categories, respectively. While ManipLLM often struggles to accurately locate the movable part, A3VLM, despite its strong visual grounding ability, often fails to translate its object-centric representation into the fine-grained affordance to guide precise actions. In contrast, by leveraging the commonsense knowledge from foundation models, our method can select the suitable manipulation concept and accurately localizes the actionable part for each object. Then the prior knowledge embedded within the selected concept can be grounded onto the target part via parameter estimation. These process allows our method to generate physically-grounded, precise actions, directly contributing to our higher success rates and enhanced generalization across categories.

#### 4.1.3 SYSTEM ERROR BREAKDOWN

To evaluate the bottleneck of our pipeline and their impact on manipulation performance, we conduct a system error breakdown analysis as shown in Table 2. Our approach is to sequentially provide ground-truth information for each stage, as the ground-truth data for a subsequent process relies on the output of the preceding stage. This methodology allows us to precisely isolate and analyze the contribution of each module.

Our initial ablation on the concept selection and actionable part grounding modules reveals them to be the primary bottleneck. Providing ground-truth concept selection leads to a modest improvement in the success rate of 65%, highlighting the importance of correctly mapping a high-level instruction to a proper manipulation concept. The success rate then sees a more significant jump of additional 10 percentage points to 75% with ground-truth actionable part grounding. This indicates that the core challenge lies in the upstream perception modules' ability to provide accurate, part-level in-

Table 2: System error breakdown. Results are reported as average success rates. "✓" indicates the use of ground-truth, otherwise estimated. Ground truths are obtained following the data collection phase.

| Setting | Components | | | | | Success Rate |
|---|---|---|---|---|---|---|
| | CS | APG | STRU | POSE | AFF | |
| Baseline | - | - | - | - | - | 0.59 |
| + Concept Selection (CS) | ✓ | - | - | - | - | 0.65 |
| + Actionable Part Grounding (APG) | ✓ | ✓ | - | - | - | 0.75 |
| + Structural Parameters (STRU) | ✓ | ✓ | ✓ | - | - | 0.78 |
| + Pose Parameters (POSE) | ✓ | ✓ | ✓ | ✓ | - | 0.83 |
| + Affordance Parameters (AFF) | ✓ | ✓ | ✓ | ✓ | ✓ | 0.89 |

Table 3: Performance comparison with baselines using different end-effectors. Our method demonstrates a more significant advantage with the parallel gripper than with the suction gripper, due to the higher manipulation difficulty of the former.

| Method | Suction Gripper | | Parallel Gripper | |
|---|---|---|---|---|
| | Train Cat. Acc. | Test Cat. Acc. | Train Cat. Acc. | Test Cat. Acc. |
| ManipLLM | 0.61 | 0.56 | 0.40 | 0.37 |
| A3VLM | **0.79** | 0.75 | 0.46 | 0.40 |
| **Ours** | 0.78 | **0.81** | **0.59** | **0.58** |

formation. This bottleneck primarily stems from the inherent limitations of the vision foundation models and the simulated environment. Specifically, Grounding DINO struggles to accurately detect object parts due to its training on object-level data (Liu et al., 2024). Concurrently, VLMs face challenges from the lack of visual realism and depth cues in the SAPIEN (Xiang et al., 2020) simulator's rendered images, which impedes their ability to accurately identify and reason about objects.

To evaluate the impact of parameter estimation, we further provide ground-truth parameters for the subsequent stages, including geometric structure, 6-DoF pose, and affordance parameters. As we progressively provide ground-truth for each of these parameters, the success rate steadily climbs, ultimately reaching a high score of 89% under these ideal conditions. This result strongly demonstrates that once the input parameters for a manipulation concept can be precisely acquired, the embedded manipulation knowledge can efficiently and accurately guide the robot to complete tasks.

### 4.1.4 MANIPULATION CONCEPT WITH SUCTION

We also conduct experiments using a suction gripper as the end-effector to demonstrate the versatility of our proposed manipulation concept. This shows that manipulation concepts can be defined to describe not only gripper-based skills but also suction-based ones. This setting also enables a fairer comparison with the ManipLLM (Li et al., 2024) method, which is specifically designed for suction, unlike the A3VLM (Huang et al., a) approach that is agnostic to the end-effector type. Our experimental results, presented in Table 3, show a notable increase in success rate when using a suction gripper compared to a parallel gripper. This is likely because the parallel gripper, being less robust to variations in pose and contact, presents a higher manipulation difficulty, which is reflected in its lower success rate. In principle, manipulation concept can be extended to diverse end-effectors; in the Appendix A.5.1, we illustrate its extention to a dexterous hand, including simulation demonstrations.

## 4.2 REAL-WORLD EVALUATION

### 4.2.1 EXPERIMENT SETUP

To further validate our method's robustness and generalizability, we conduct a series of experiments on various real-world household objects. Our setup utilizes a robot arm with a parallel gripper and a

Figure 4: Real-robot manipulation experiments across six articulated objects. Each **column** shows one object category, with images from **top to down**: (1) the object used for evaluation, (2) robot initial grasp, (3) manipulation completion. Tasks (from left to right): bottle opening, box opening, bucket handle rotation, cabinet drawer opening, laptop opening, and pot lid lifting.

RealSense 415 camera for capturing RGB-D data. We evaluate our framework's performance on six categories of articulated objects, each with a corresponding manipulation task. These tasks include *opening a bottle, opening a box, rotating the handle of a bucket, lifting the cover of a pot, opening a laptop, and opening the drawer of a cabinet*, as shown in Fig. 4. We adopt success rate as our evaluation metric. More experimental details can be found in the Appendix A.3.

### 4.2.2 RESULTS AND DISCUSSION

The results of real-world experiments are shown in Table 4, demonstrating our method's efficacy in practical settings. Unlike approaches that rely on the visual appearance of entire objects (Mo et al., 2021; Li et al., 2024; Huang et al., a), our method focuses on the geometric shape and affordance of task-relevant parts. This design is crucial for enabling robust generalization across intra-class variations and diverse scenarios. Moreover, this approach effectively mitigates the common sim-to-real gap issues caused by discrepancies in texture, lighting, and material, which is an essential capability for reliable real-world robotic systems.

Table 4: Real world experiments.

| Object | | | | | | |
|--------|-------|------|------|-------|------|-------|
| Ours | 10/10 | 9/10 | 8/10 | 10/10 | 9/10 | 10/10 |

## 5 CONCLUSION

In this paper, we introduced a novel representation termed **Manipulation Concept**, which encodes prior knowledge about manipulation skills into a set of parameterized templates in the form of code. Building upon this powerful representation, we developed a comprehensive framework for generalizable and physically-grounded manipulation of articulated objects. Our framework leverages a visual-language model for task-relevant information extraction and concept selection, then uses the robustly estimated parameters to instantiate the selected concept, grounding abstract knowledge into precise robotic actions. Experimental results in both simulation and real-world environments demonstrate the effectiveness and superiority of our approach. This work paves a new way for achieving more generalizable and robust robotic manipulation skills. Future work could focus on extending our framework to handle complex tasks involving multi-object manipulation.

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

# A APPENDIX

This appendix is organized as follows.

- In Sec. A.1 we provide method details about the definition of Manipulation Concept (A.1.1), each module of the framework (A.1.2-A.1.5), and the data collection process (A.1.6).
- In Sec. A.2 we provide details about simulation experiments, including the setting, and more experiments.
- In Sec. A.3 we provide details about real-world experiments.
- In Sec. A.4 we provide the meaning of object category icons in the main paper.
- In Sec. A.5 we provide additional discussion about the extensibility of manipulation concept to diverse end-effectors (A.5.1), the comparison between our manipulation concept and other representations (A.5.2), how our method deals with irregular geometries (A.5.3), the scalability of manipulation concept creation (A.5.4), the robustness analysis of pose estimation module (A.5.5), and future work (A.5.6).
- In Sec. A.6 we provide code example of manipulation concept to better illustrate this representation.

## A.1 METHOD DETAILS

### A.1.1 THE DEFINITION OF MANIPULATION CONCEPT

To cover the manipulation tasks for various articulated object categories, we curate a concise set of actionable part types. We consider four fundamental geometric primitives: *Cuboid*, *Ring*, *Cylinder*, and *Sphere*. For each primitive, we associate a set of semantic categories that commonly appear in real-world objects. Each (geometry, semantic) pair corresponds to a distinct part type in our manipulation concept definition. This design allows our manipulation concepts to be reusable across object instances sharing similar shape and function.

The mapping is defined as follows:

- **Cuboid**: Handle, Door, Cover, Button, Window, Seat
- **Ring**: Handle, Door, Spout, Cover
- **Cylinder**: Cover, Handle, Button
- **Sphere**: Handle

### A.1.2 CONCEPT SELECTION

To extract task-relevant information, we choose GLM-4.1V-9B-Thinking (Hong et al., 2025) as our vision-language model, as its powerful visual reasoning capabilities are sufficient for our tasks. To ensure the information extraction process is robust and accurate, we employ a progressive, hierarchical approach. At each step, we query the VLM with a targeted question and a constrained set of candidates based on the previous step's response. In this way, we sequentially acquire all three types of information, including object category, part type, and action type. The specific prompts used for task-relevant information extraction are detailed in the table 5.

### A.1.3 ACTIONABLE PART GROUNDING

To capture part-level point clouds, we use Grounded-SAM (Ren et al., 2024) that combines Grounding DINO (Liu et al., 2024) and the Segment Anything Model (SAM) (Kirillov et al., 2023). Grounding DINO (Liu et al., 2024) serves as an open-set object detector, which is used to generate bounding box prompts for SAM (Kirillov et al., 2023). For certain object categories, directly prompting Grounding DINO (Liu et al., 2024) with the part name does not yield optimal detection results. To address this, we construct specific prompts for each combination of object category and part to achieve a better result.

Table 5: Prompts used for task-relevant information extraction

| Prompt ID | Description and Example |
|---|---|
| 1. object_category | This prompt is used to predict the object's category from a predefined set.

*Example: "Given an image of an object, please identify its category from the following set: [list of categories]."*
*Output format:* object category: <identified category> |
| 2. part_type | This prompt is used to identify the part to manipulate, given the object category and task. It provides a set of candidate part types, which are determined by the identified object category, each paired with a natural language description of its appearance and function.

*Example: "Given an image of a [object category] and the task '[task]', please select the most suitable part to interact with from the following options:*
*– [part_type_1]: [Description of the part with functional and structural cues.]*
*– [part_type_2]: [Another description of the part.]*
*Output format:* part type: <selected part> |
| 3. action_type | This prompt is used to select the appropriate action, given the object category, part type and task. It provides a set of candidate action types, which are determined by the selected part type, each paired with a natural language description of its motion semantics and interaction direction.

*Example: "Given the object category [object category], part type [part type] and the task '[task]', please select the most suitable action from the following options:*
*– [action_type_1]: [Description of the action of how it interacts with the part.]*
*– [action_type_2]: [Another description of the action.]*
*Output format:* action type: <selected action> |

Since SAM (Kirillov et al., 2023)'s segmentation of parts is not always precise, we perform a post-processing step on the resulting point cloud. We use the DBSCAN algorithm (Schubert et al., 2017) to filter out outliers, ensuring a cleaner and more accurate representation of the object part.

### A.1.4 GEOMETRIC PARAMETER ESTIMATION

**Structural Parameters.** A key challenge in predicting parameters for 3D geometric primitives like cuboids and cylinders is their inherent symmetry. For instance, a single cuboid can be described by multiple valid parameter sets (e.g., swapping the values for length and width results in the same shape for a cuboid), which creates ambiguity in the supervision signal and can hinder model training.

To address this, we standardize the point cloud of each primitive to a canonical pose. For a cuboid-shaped part, we first obtain the pose of the 3D bounding box, which approximates the part's pose. We then use the rotation ($R$) and translation ($T$) components of this bounding box to transform the part's point cloud. By doing so, we ensure a unique prediction target for its length, width, and height, which are aligned with the $x$, $y$, and $z$ axes, respectively. A similar standardization process is applied to cylinders and spheres.

Additionally, to enhance the model's robustness and generalization, we normalize the point cloud to fit within the unit ball. We also apply random rotational perturbations as a form of data augmen-

tation. These rotations are generated from a normal distribution with a mean of 0 and a standard deviation of 0.4, ensuring the model can handle slight variations in orientation.

**6-DoF Pose Parameters**   To obtain the object's pose, we leverage FoundationPose (Wen et al., 2024), a pose estimation method that can operate in both model-based and model-free settings. It works by taking a segmentation mask to extract the target part's point cloud from the scene and then estimates the pose of the part's corresponding CAD model within this captured point cloud. This pose is then transformed from the camera's frame to the world coordinate system.

Given the inherent symmetries of geometric primitives like cuboids and cylinders, the initial pose from FoundationPose (Wen et al., 2024) can be ambiguous. We therefore apply a post-processing step to poses based on prior knowledge about the part, such as the predicted concept, object category, and surrounding point cloud information, to obtain a precise and unambiguous representation of the part's geometric parameters.

For cuboid primitives, such as those representing doors, handles, or covers, we perform the following post-processing to obtain a consistent and unambiguous pose: (1) We first determine the object's Z-axis as the direction of its smallest dimension. (2) The Y-axis is then defined as the direction with the smallest angle relative to the upward vector. (3) The remaining axis defines the X-axis, with its positive direction determined by our initial placement assumption. For handles, we use the RANSAC algorithm to estimate the plane on which the part lies, which helps in further refining its orientation. A similar process is applied to other primitives. In our experiments, objects were initially placed such that their front side generally faced the camera. This practical setup ensures the manipulable parts are readily accessible to the robot's end-effector, which also helps the post-processing of estimated pose.

### A.1.5 AFFORDANCE PARAMETER ESTIMATION

For affordance parameter estimation, during the training phase, the model predicts the weights, means and covariance matrices of the Gaussian Mixture Model (GMM). In the inference phase, we sample from the output distribution modeled by the GMM. The estimated manipulation parameters are then used to instantiate the manipulation concept, analytically computing the precise grasp pose and force direction. The loss function during training is as follows:

$$\mathcal{L}_{\text{GMM}} = -\log p(\mathbf{y} \mid \boldsymbol{\theta}) = -\log \left( \sum_{k=1}^{K} \alpha_k \cdot \mathcal{N}(\mathbf{y} \mid \boldsymbol{\mu}_k, \boldsymbol{\Sigma}_k) \right)$$

During inference, we sample from the following distribution:

$$p(\mathbf{y} \mid \boldsymbol{\theta}) = \sum_{k=1}^{K} \alpha_k \cdot \mathcal{N}(\mathbf{y} \mid \boldsymbol{\mu}_k, \boldsymbol{\Sigma}_k)$$

where $K$ is the number of mixture components, $\alpha_k$ are mixing weights, and $\mathcal{N}(\mathbf{y} \mid \boldsymbol{\mu}_k, \boldsymbol{\Sigma}_k)$ denotes the $k$-th Gaussian component with mean $\boldsymbol{\mu}_k$ and covariance $\boldsymbol{\Sigma}_k$.

### A.1.6 DATA COLLECTION

In the ConceptFactory dataset (Sun et al., 2024), each object is annotated with a set of analytic concepts and their corresponding parameters. However, the annotations' structure is often composed of multiple parts, which in turn consist of multiple geometric primitives. For example, a "window" concept might encompass several windows, with each one comprising multiple cuboids. This nested structure is inconsistent with the single part-parameter format required by our approach. To address this, we develop a programmatic translation process to convert these annotations into our chosen set of geometric primitives, ensuring the converted shapes closely approximate the original geometry.

The ConceptFactory (Sun et al., 2024) dataset provides annotations for objects in their rest state. To leverage this static information for objects in random states, we first establish a crucial mapping: at the rest state, we programmatically record which part corresponds to a specific link ID and visual ID provided in SAPIEN simulator (Xiang et al., 2020). This mapping serves as a lookup table for all subsequent operations.

During a task when the object is in a random configuration, we use this pre-computed mapping to obtain the necessary data. To acquire the point cloud of a specific part, we use its corresponding visual ID to query the SAPIEN simulator (Xiang et al., 2020). To acquire the part's pose, which is essential for our affordance parameter estimation, we obtain the current joint angle from the mapped link. We then use this angle to calculate a transformation matrix that gives us the accurate pose of the part. The data collection procedure follows the setting detailed in Sec. 4.1.1.

To address inconsistencies between the visual and collision shapes in the original PartNet-Mobility (Xiang et al., 2020) URDF files (*e.g.*, a door that has a visual handle but lacks a corresponding collision mesh) we leverage a modified version of the dataset provided by Where2Act (Mo et al., 2021). This dataset is pre-processed using the Voxelized Hierarchical Approximate Convex Decomposition (VHACD) algorithm (Mamou et al., 2016), which provides more accurate collision meshes. For object categories included in Where2Act (Mo et al., 2021), we use their provided data. For all other categories, whose visual and collision shapes do not differ significantly, we use the original PartNet-Mobility (Xiang et al., 2020) data. This same processed dataset is also used during our experiment phase.

## A.2 SIMULATION EXPERIMENT

### A.2.1 EXPERIMENT SETTING

The camera is positioned in the upper hemisphere, looking at the center of the scene. Its position is randomized for each experiment, with a random azimuth in the range of $[120°, 240°)$ and a random elevation in the range of $[30°, 60°]$. The camera's distance to the object is also randomized, falling uniformly within $[4.5, 5.5]$ units.

For the initial state of the object's articulated joints, we use a hybrid approach. With a probability of $0.8$, we set the joints to a "random-middle" state, and with a probability of $0.2$, we set them to a "closed" state. To ensure that the "random-middle" state does not degenerate to the "closed" state due to gravity, we disable gravity for objects belonging to specific categories. The object's scale is set to maintain a realistic size ratio relative to the gripper.

**Evaluation Metric.** An interaction is considered successful if the movable part of the object satisfies two conditions: First, its movement distance must be significant, either exceeding a predefined absolute threshold of 0.03 meters or a relative threshold of 50% of its total motion range. Second, its motion must be consistent with the intended direction. This consistency is quantitatively measured by the dot product of the motion vector and the intended direction vector, requiring a value greater than 0.3.

### A.2.2 A3VLM BASELINE REPLICATION

Due to the manipulation policy module of A3VLM (Huang et al., a) not being fully open-sourced at the time of this paper's submission, we mainly follow the methodology described in their paper to replicate the A3VLM (Huang et al., a) baseline.

Our implementation consists of two stages. In the first stage, we query the model with the instruction: "Please execute the task described with 3D rotated bounding box representations by the following instruction: [Action] the [category]." From this, we obtain the predicted action type and the 3D bounding box (BBox) of the relevant object part. In the second stage, we prompt the model again with: "Please provide the joint's type and its 3D axis linked to the object part: BBox B." This yields the joint type and the corresponding rotation axis associated with the part.

With the bounding box, joint type, rotation axis, and action type all obtained, we then determine the grasp pose. We employ GraspNet (Fang et al., 2020) to generate multiple candidate grasps, selecting the highest-scoring grasp pose that lies within the predicted bounding box, instead of randomly selecting a grasp pose as suggested in the original paper. Following the original A3VLM (Huang et al., a) strategy, we apply a heuristic method to construct the manipulation trajectory based on the estimated bounding box, joint type, rotation axis, and action type:

- For 'scroll' actions: We adhere to the original paper's method, ensuring the grasp pose overlaps with the rotation axis A.

- For 'Slide' actions: We have the gripper move along the axis.
- For 'Rotate' actions: We move the gripper perpendicularly to the plane formed by the rotation axis and the line connecting the center of bounding box to the rotation axis.

This replication strategy ensures a fair comparison with the original A3VLM methodology.

### A.2.3 INFERENCE SPEED ANALYSIS

We measure the inference time of each stage in our pipeline and compare the total latency against two strong baselines: ManipLLM (Li et al., 2024) and A3VLM (Huang et al., a). The total inference time, defined as the duration from raw RGB-D image input to the initial execution of the final 7-DoF grasp pose and force direction, is measured on a single NVIDIA RTX 3090 GPU. All reported latencies are averaged over 10 repeated runs under identical conditions. As shown in Table 6, the VLM-based concept selection takes 3.19 s. Part grounding requires 1.55 s, followed by geometric parameter estimation (1.70 s) and affordance parameter estimation (0.44 s), while template instantiation is negligible. The overall pipeline latency is approximately 7 seconds. Notably, our approach is as fast as the baseline ManipLLM (6.97s vs. 6.72s) and significantly faster than the best-performing baseline A3VLM (60.10s).

Table 6: Per-stage inference latency (in seconds) on an NVIDIA RTX 3090 GPU. All values are averaged over 10 runs.

| Phase | Stage | Average Latency (s) |
|---|---|---|
| **Initialization** | Concept Selection (VLM-based) | 3.19 |
| | Part Grounding & Segmentation | 1.55 |
| | Geometric Parameter Estimation | 1.70 |
| | Affordance Parameter Estimation | 0.44 |
| **Execution** | **Exection Phase Latency** | 0.09 |
| **Total** | **Total Latency** | 6.97 |
| **Total** | **ManipLLM (Li et al., 2024)** (baseline) | 6.72 |
| **Total** | **A3VLM (Huang et al., a)** (baseline) | 60.10 |

### A.3 REAL-WORLD EXPERIMENTS DETAILS

**Video demonstrations are shown in the supplementary video**. For each object, we randomize its initial pose and joint state. Each task is tested for ten trials and we adopt success rate as a metric. Experiments on the bottle and pot were conducted in cluttered background environments, demonstrating the generalization capability of our method. Figure 4 shows the six objects used in our real-robot experiments and their corresponding manipulation scenarios.

Manipulation concept is inherently gripper-aware, which means it require specific information about the end-effector, such as finger length. When we transition from simulation to a physical robot, we must update this information to match the data of the gripper currently in use.

While a simulation environment only requires considering a gripper's pose, real-world deployment necessitates accounting for the full robot arm's joint space. We found that manipulating objects like cabinet and bucket could sometimes exceed the robot's joint limits, a common challenge in physical robotics. To ensure the output actions are within the robot's capabilities, we apply a post-processing step to the gripper's pose, adjusting it to remain within the robot's reachable workspace. For instance, because a cylindrical cover is symmetrical about its Y-axis, its pose has inherent ambiguities that lead to a multi-solution space for valid grasp poses. We then select the most feasible pose that lies within the robot's reachable workspace. We also set the initial poses of the objects to a reasonable, reachable configuration.

For these experiments, we focus on evaluating the accuracy of our parameter estimation module and the knowledge encoded within the manipulation concepts themselves. Therefore, we directly

Table 7: Object categories with corresponding icons

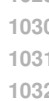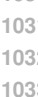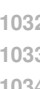

| Bottle | Box | Bucket | Door | Dishwasher | Display | Faucet | Foldinchair | Kettle | KitchenPot | Laptop | Microwave |
|---|---|---|---|---|---|---|---|---|---|---|---|

| Oven | Pen | Refrigerator | Safe | Stapler | Switch | StorageFurniture | Table | Trashcan | USB | Washingmachine | Window |
|---|---|---|---|---|---|---|---|---|---|---|---|

provide the ground truth for the object category and the bounding box of the target part, allowing us to evaluate our core method.

### A.4 REPRESENTATION FOR EACH CATEGORY ICON

In Table 7, we provide an overview of the meaning of each category icon in Table 1 in the main paper. These categories, along with their corresponding objects, are sourced from PartNet-Mobility (Xiang et al., 2020)

### A.5 ADDITIONAL DISCUSSIONS

#### A.5.1 EXTENSIBILITY OF MANIPULATION CONCEPT TO DIVERSE END-EFFECTOR TYPES

The primary objective of this paper is the introduction and validation of the manipulation concept and corresponding framework. To enable efficient validation of the manipulation concept's feasibility and the framework's performance, we focused our experiments on the most common and basic end-effectors, namely the parallel and suction grippers.

In principle, the manipulation concept is extensible to various types of end-effectors (*e.g.*, multi-fingered hands). The core idea of the manipulation concept is that the concept models the physics-informed interaction between the object and the end-effector through analytic spatial transformations, while simultaneously considering their structural properties. Consequently, the end-effector's specific characteristics can be explicitly encoded in the manipulation concept's definition for gripper-part interaction.

For instance, consider the "*pull the ring handle with full hand*" concept implemented on a 5-fingered dexterous *Shadow Hand*. The design of this new concept is summarized below:

- Initial Hand Pose: The hand's root pose is initialized such that the palm faces the ring, with the palm normal aligned parallel to the ring's central axis, and the fingertips aligned parallel to the ring's axial direction.
- Fingers Initial Configuration: For the four fingers (excluding the thumb), the adduction/abduction joint (sideways movement) is set to 0, ensuring the fingers are laterally aligned. Based on the ring's axial thickness and radial thickness, a collective bending angle is calculated and proportionally distributed across the joints to conform the four fingers to the ring's surface. The thumb's metacarpophalangeal joint (the second joint from the fingertip) is set to 0.8, allowing the thumb to be parallel to the other four fingers and make contact with the ring's surface.
- Affordance Parameters: We design seven sets of affordance parameters, categorized by their control function: (a): the abduction/adduction angle (sideways movement) in the four fingers, (b): the bending angle of the distal (fingertip) and middle joints (the two joints near the fingertip), (c): the bending angles of the proximal joints (closest to the palm), (d): the rotation control of the thumb along the ring's surface, (e)-(g) palm's relative translation and rotation with respect to the ring (consistent with the parallel gripper).
- Force direction: The required force direction for the dexterous hand aligns with that of the parallel gripper.

As strong evidence of our method's extensibility, we further evaluate this design in a simulation of the *"Open the door of a microwave"* task. The visualization of this concept design, along with the task execution, is presented in Figure 5.

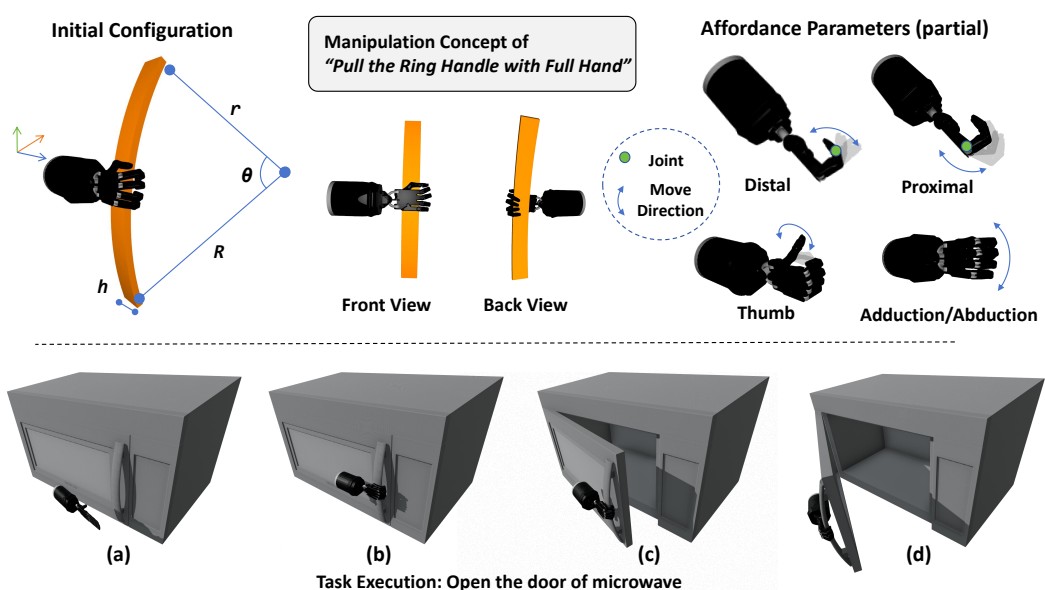

Figure 5: The visualization of the concept design for a 5-fingered dexterous hand and its execution of the microwave door-opening task.

### A.5.2 COMPARISON WITH PRIOR STRUCTURED REPRESENTATIONS

Manipulation concept is a novel representation that differs from existing structured object representations in three key aspects:

1. **Part representation.** Manipulation concepts represent actionable parts using geometric primitives (e.g., cylinders, cuboids), explicitly encoding their geometric shapes to provide more actionable information. In contrast, prior part-level representations, such as those in GAPartNet (Geng et al., 2023b) and A3VLM (Huang et al., a), typically model parts using representation like 6D poses or 3D bounding boxes, which lack detailed shape information necessary for precise interaction.

2. **Explicit gripper-awareness.** Manipulation concepts jointly model the actionable part and the end-effector, explicitly incorporating gripper characteristics (e.g., kinematic structure, opening width) into the action derivation. This ensures that generated actions physically adhere to the gripper's physical constraints. In contrast, most prior representations derive manipulation policies using only object-centric information, without accounting for the physical properties of the gripper.

3. **Learning mechanism in interaction strategy.** The design of manipulation concepts enables a learning-based approach to predict affordance parameters that directly govern the gripper's interaction pattern (e.g., grasp pose), supporting flexible and adaptive manipulation. Prior methods typically rely on heuristic interaction strategies, limiting their adaptability across diverse tasks and objects.

Together, these novel designs enable manipulation concepts to produce actions that are more precise, physically feasible, and generalizable across object categories.

### A.5.3 ROBUSTNESS WHEN DEALING WITH IRREGULAR GEOMETRIES

In our work, we use parameterized geometric primitives to describe the geometric structure of the actionable part. However, some parts exhibit irregular geometries that cannot be accurately approximated by standard primitives. Here, we explain how our method robustly handles such cases.

As discussed in Sec. 3.1, manipulation concept encodes the geometric structure (possibly not well approximated) to indicate affordance knowledge (e.g., how to interact with a part), which defines a

feasible interaction space. Then this affordance knowledge is grounded via a learning mechanism that estimates the affordance parameters. The learning module uses the actionable part's point cloud (which may possess irregular geometry) as input, enabling our method to inherently account for geometric details and derive precise gripper actions that adaptively handle local geometric variations, thereby alleviating the gap in representing irregular geometries.

In this way, our approach achieves robust manipulation even for parts with complex or irregular shapes. This is validated by the high success rates across our manipulation tasks on PartNet-Mobility (Xiang et al., 2020), a dataset comprising diverse articulated objects with a wide range of geometric characteristics (including many with irregular structures).

### A.5.4 SCALABILITY OF MANIPULATION CONCEPT CREATION

Manually constructing manipulation concepts does not pose a critical bottleneck to scalability. This conclusion is based on three key factors:

1. **Broad concept coverage minimizes the need for crafting new concepts.** Diverse real-world objects often share geometrically and functionally similar parts; therefore a single concept can offer broad coverage across categories (e.g., "Ring_Handle_Pull" on bucket, kettle, kitchen pot, etc.). Consequently, extending to a new object category requires creating only a few (sometimes even zero) additional concepts. This high generality significantly reduces the workload required to extend the concept to a new object category.

2. **Crafting new concepts requires minimal labor.** Because of (1) **Code reusability mechanisms**: Since manipulation concepts are implemented as programmatic templates, hand-crafting them is efficient in practice: new concepts can be conveniently constructed by inheriting from and composing reusable functions and classes. For instance, a function for *"grasping a ring"* can be reused across various concepts, significantly boosting development efficiency. (2) **Visualization tool for manipulation knowledge:** We have developed a visualization tool that provides an interactive display of the manipulation knowledge for each concept, as shown in Fig. 6 to aid in its construction, making the manual definition process more efficient and accurate.

3. **Crafting new concepts is relatively scalable.** It is inevitable for articulated object manipulation methods (Geng et al., 2023b; Li et al., 2024; Huang et al., a; Mo et al., 2021; Geng et al.) to collect training data (involving object collection, annotation, and data cleaning, etc.) for optimal performance on additional categories due to the structural differences between categories. As discussed above, crafting new concepts requires minimal labor, typically consuming less than 1/10 of the time allocated for data collection. Therefore, compared with truly labor-intensive data collection, the labor of crafting new concepts is negligible and poses no obstacle to scaling to additional categories.

To further validate the efficiency of this process, we recruited three undergraduate volunteers, each of whom independently crafted one concept for each of the four novel categories (scissors, lighter, pliers, mouse), and we verified the effectiveness of their manipulation knowledge in experiments. We found that, with the aid of the visualization tool, a new concept could be completed in 2.1 hours on average, as it primarily involves common-sense knowledge and basic spatial transformations. This practical evidence strongly validates the scalability of manipulation concept creation process. Furthermore, we will open-source the constructed manipulation concepts to reduce labor costs and promote broader research in the community. Future work will investigate automating the construction of the concept library using AI agents, as well as enabling VLMs to generate executable manipulation concept code on-the-fly during inference based on task instructions.

### A.5.5 ROBUSTNESS ANALYSIS OF POSE ESTIMATION MODULE

We use parameterized geometric primitives (e.g., cylinders, cuboids) estimated from the point clouds as CAD models for FoundationPose (Wen et al., 2024). These primitives lack fine geometric details. To analyze the robustness of FoundationPose when using such geometric primitives as CAD models, we conduct a case study evaluating its performance when the real object exhibits additional geometry beyond the primitive. As illustrated in Figure 7, even when the target part's point cloud contains extra geometric details (e.g., protrusions or attachments), FoundationPose remains robust enough

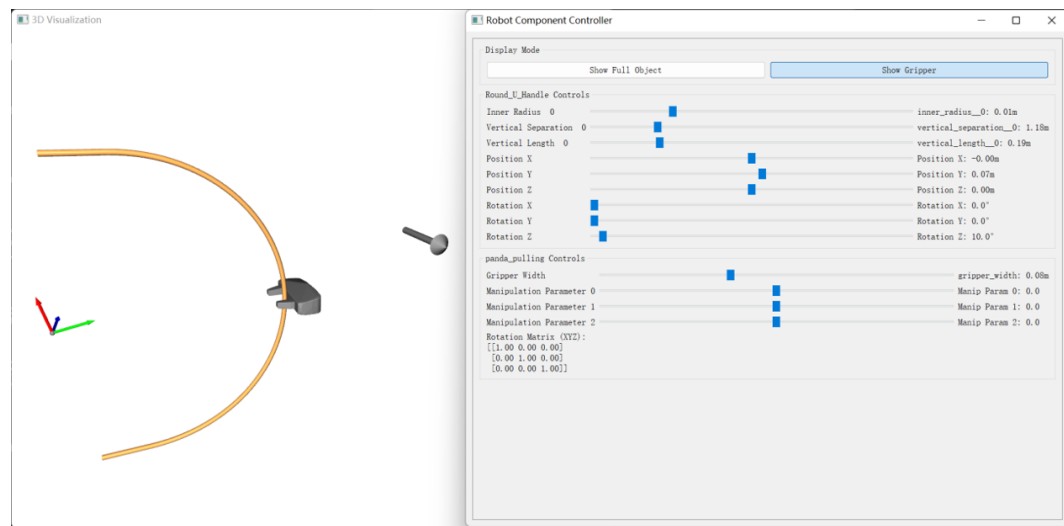

Figure 6: Visualization tool for creating the manipulation concept.

to accurately predict the pose of the simplified primitive, achieving reasonable alignment with the actual part's point cloud. This alignment allows the robot to complete the manipulation task with a high success rate. Across the majority of our experiments, we found that the impact of pose error on the manipulation success rate remains within an acceptable tolerance.

To obtain a geometrically more accurate CAD mesh, we explore using CAD recovery techniques to reconstruct detailed part-level meshes, which we then use as the CAD model for FoundationPose. We then compare them with our primitive-based method. Specifically, we attempt to reconstruct the part-level mesh using a technique in OnePoseViaGen (Geng et al., 2025). The comparison results are presented in Table 8, which shows that both approaches achieve comparable performance. This further demonstrates that the primitives are sufficient for robust pose estimation in our setting.

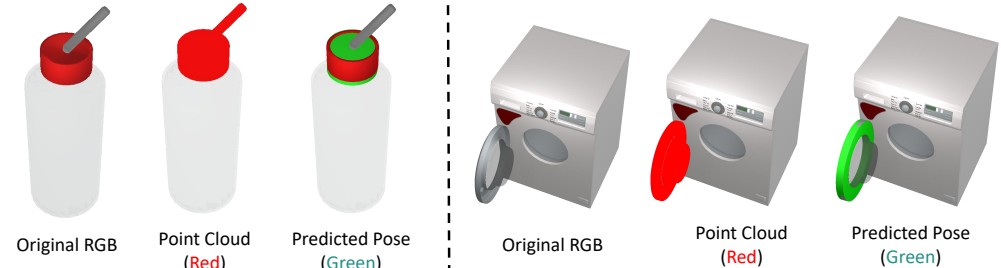

Figure 7: The robustness of FoundationPose when dealing with parts featuring extra geometry. The left and right part respectively show the pose estimation results for the cylindrical lid of a bottle and the ring-shaped door of a washing machine. In each section, from left to right, we display: the original RGB image, the point cloud used for pose estimation (in red), and the estimated primitive after pose estimation (in green).

Table 8: Comparison of ADD-S metric between learned CAD recovery and our primitive-based approach for pose estimation.

| Method | ADD-S ↓ |
|---|---|
| CAD Recovery | 0.055 |
| Ours (Primitive-based) | 0.058 |

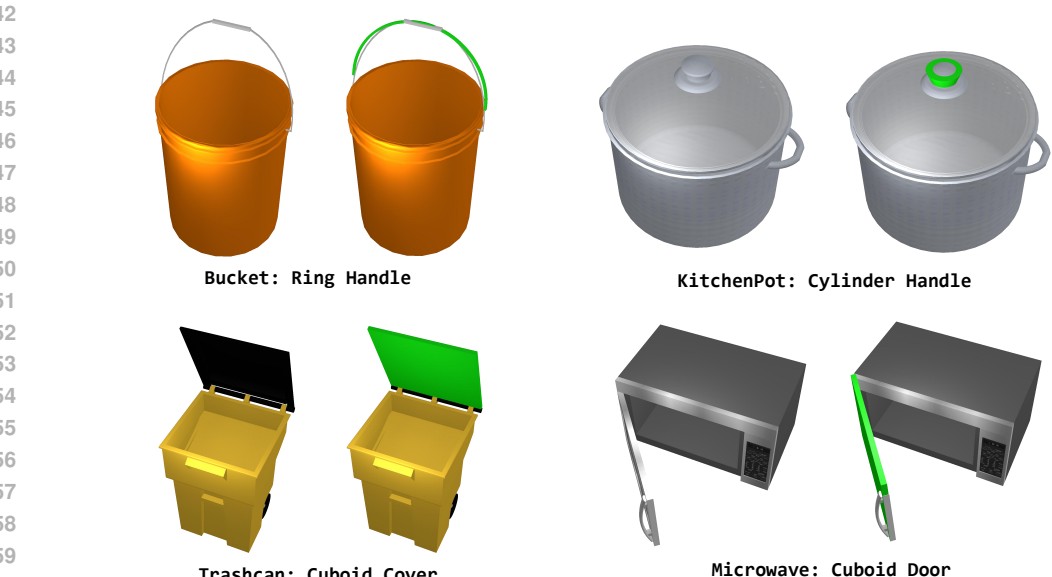

**Bucket: Ring Handle**        **KitchenPot: Cylinder Handle**

**Trashcan: Cuboid Cover**        **Microwave: Cuboid Door**

Figure 8: Manipulation concepts cover a wide range of object categories in the PartNet-Mobility dataset.

### A.5.6 LIMITATIONS AND FUTURE WORK

Our current approach focuses only on gripper-based manipulation of rigid articulated objects represented with simple geometric primitives. It does not handle deformable objects, dynamic interactions, or complex end-effectors such as dexterous hands. In the future, we will extend the design of the Manipulation Concept to support a broader range of end-effectors, including dexterous hands or tool use, by explicitly modeling their specific characteristics. Concurrently, we aim to enhance the geometric representation of object parts beyond simple primitives to better capture complex part geometries. For deformable objects manipulation, we will develop a new parameterized representation that expresses the shape of a deformable object as a parameterized transformation (e.g., folding or bending) from a canonical base configuration (e.g., a flat cloth or straight cable). Additionally, we will explore complex manipulation tasks involving multiple rigid objects, force control, dynamic interactions, and long-horizon task execution by encoding more knowledge into the manipulation concept. For instance, for tasks requiring precise force control or dynamic responses, we can incorporate time, contact forces, object velocities, and detailed spatial context as input parameters, and further predict force and velocity as part of the output actions.

### A.6 CODE EXAMPLE OF MANIPULATION CONCEPT

Here we present an implementation example of a Manipulation Concept, using ***Pull_Ring_Handle*** to illustrate the design. This concept encodes the skill of a parallel gripper lifting a ring-shaped handle.

We first define a reusable Ring Primitive class, whose parameters control the ring's geometric size and pose. Building upon this, we implement the ***Pull_Ring_Handle*** concept, which analytically computes a 7-DoF grasp pose and force direction from geometric and affordance parameters. Its core function, ***grasp_ring***, is designed for reuse across other manipulation concepts. Specifically, the grasp pose is initialized based on the ring's geometric parameters, then refined by an offset derived from the affordance parameters in the ring's local frame. The resulting pose is finally transformed into the world coordinate system.

```
class Ring(GeometryTemplate):
    def __init__(self, height, outer_radius, inner_radius, exist_angle=np
    .pi * 2,
```

```
                    position=[0, 0, 0], rotation=[0, 0, 0], rotation_order="
XYZ"):
    """
    Ring primitive with circular cross-section and straight sides.

    :param height: Height along Y-axis.
    :param outer_radius: Outer radius in X-Z plane.
    :param inner_radius: Inner radius in X-Z plane.
    :param exist_angle: Angular span of the ring (default: full
circle 2 pi).
    :param position: Global position (x, y, z).
    :param rotation: Euler rotation angles (x, y, z).
    :param rotation_order: Rotation order (e.g., "XYZ").
    """
    super().__init__(position, rotation, rotation_order)

    # Store parameters
    self.height = height
    self.outer_radius = outer_radius
    self.inner_radius = inner_radius
    self.exist_angle = exist_angle
    self.position = position
    self.rotation = rotation
    self.rotation_order = rotation_order

    # Generate vertices and faces
    num_segments = 256
    vertices = []
    faces = []

    half_height = height / 2.0

    for i in range(num_segments + 1):
        theta = exist_angle * i / num_segments

        cos_t = np.cos(theta)
        sin_t = np.sin(theta)

        # Outer top/bottom
        outer_top = [outer_radius * cos_t,  half_height, outer_radius
 * sin_t]
        outer_bottom = [outer_radius * cos_t, -half_height,
outer_radius * sin_t]

        # Inner top/bottom
        inner_top = [inner_radius * cos_t,  half_height, inner_radius
 * sin_t]
        inner_bottom = [inner_radius * cos_t, -half_height,
inner_radius * sin_t]

        vertices.extend([outer_top, outer_bottom, inner_top,
inner_bottom])

    # Build faces
    for i in range(num_segments):
        v0 = 4 * i
        v1 = v0 + 1
        v2 = v0 + 2
        v3 = v0 + 3
        v4 = v0 + 4
        v5 = v0 + 5
        v6 = v0 + 6
        v7 = v0 + 7

        # Side faces (outer and inner walls)
```

```
1350              faces.append([v0, v1, v5])  # outer side
1351              faces.append([v0, v5, v4])
1352
1353              faces.append([v3, v7, v2])  # inner side (note winding for
1354      correct normal)
1355              faces.append([v7, v6, v2])
1356
1357              # Top face
1358              faces.append([v0, v4, v6])
1359              faces.append([v0, v6, v2])
1360
1361              # Bottom face
1362              faces.append([v1, v3, v7])
1363              faces.append([v1, v7, v5])
1364
1365          self.vertices = np.array(vertices)
1366          self.faces = np.array(faces)
1367
1368          # Apply global transformation
1369          self.vertices = apply_transformation(self.vertices, position,
1370      rotation, rotation_order)
```

Listing 1: The code implementation of `Ring` primitive

```
class Ring_Handle_Lift(ManipulationConcept):
    def __init__(self, geometric_parameters, affordance_parameters,
    gripper_parameters):
        super().__init__(geometric_parameters, affordance_parameters,
    gripper_parameters)
        """A Manipulation Concept for lifting a ring-shaped handle with
    parallel gripper."""

        h, R, r, angle, position, rotation = geometric_parameters
        rot_x_ratio, rot_y_ratio, forward_z_ratio = affordance_parameters
        gripper_width, gripper_finger_length, gripper_palm_length,
    open_width = gripper_parameters

        self.gripper = Gripper(gripper_width, gripper_finger_length,
    gripper_palm_length, open_width)
        self.ring = Ring(h, R, r, angle, position, rotation)
        self.grasp_pose, self.force_direction = \
                self.derive_actions(rot_x_ratio, rot_y_ratio,
    forward_z_ratio)

    def derive_actions(self, rot_x_ratio, rot_y_ratio, forward_z_ratio):
        # Apply affordance parameters to adjust grasp pose
        grasp_pose = grasp_ring(self.ring, rot_x_ratio, rot_y_ratio,
    forward_z_ratio, self.gripper)
        force_direction = lifting_dir(self.ring) # upward in object frame

        # Post process the action based on structural information
        grasp_pose, force_direction = self.post_process(
        grasp_pose_, force_direction
        )
        return grasp_pose, force_direction

    def post_process(self, grasp_pose, force_direction):
        """ apply adjustment based on commonsense """

        # Ensure lifting direction is upward (physical constraint of the
    action type)
        if get_local_axis(force_direction, "y") < 0:
            force_direction = -force_direction

        return grasp_pose, force_direction
```

```python
def grasp_ring(ring, rot_x_ratio, rot_y_ratio, forward_z_ratio, gripper):
    """
    Generate a parameterized grasp pose for grasping the ring handle
    Steps:
        1. Place gripper at default grasp location.
        2. Rotate gripper around the ring's local y axis (controlled by
    rot_y_ratio).
        3. Rotate gripper around its local x-axis (controlled by
    rot_x_ratio).
        4. Shift forward along approach direction (controlled by
    forward_z_ratio).
    """
    # Default grasp at angle 0 on the ring
    grasp_position = [
            cos(ring.angle / 2) * (ring.R + gripper.palm_length),
            0,
            sin(ring.angle / 2) * (ring.R + gripper.palm_length)
        ]
    grasp_rotation = RotationMatrix('xyz', [0, -90, 0]) @ RotationMatrix(
    'y', - ring.angle)
    gripper_open_width = max(ring.h, ring.R - ring.r)
    grasp_pose = GraspPose(grasp_position, grasp_rotation,
    gripper_open_width)

    # rotate around y-axis, ring local frame
    rot_y_angle = rot_y_ratio * ring.angle / 2
    rot_y_matrix = RotationMatrix('y', rot_y_angle)
    grasp_pose = transform_pose(grasp_pose, rot_y_matrix)

    # rotate around x-axis, gripper local frame
    rot_x_angle = rot_x_ratio * pi / 2
    local_x_axis = get_local_axis(grasp_pose.rotation, "x")
    local_z_axis = get_local_axis(grasp_pose.rotation, "z")
    point_on_handle = grasp_pose.position + local_z_axis * gripper.
    palm_length
    rotate_x_matrix = rotation_around_axis(
        axis = local_x_axis,
        point = point_on_handle,
        angle = rot_x_angle
    )
    grasp_pose = transform_pose(grasp_pose, rotate_x_matrix)

    # translate along z-axis, gripper local frame
    local_z_direction = get_local_axis(grasp_pose.rotation, "z")
    forward_distance = forward_z_ratio * gripper.palm_length * 1.5
    grasp_pose.position += local_z_direction * forward_distance

    # global transformation
    obj_pos, obj_rot = self.ring.position, self.ring.rotation
    transform_matrix = TransformMatrix(obj_pos, obj_rot)
    grasp_pose = transform_pose(grasp_pose, transform_matrix)

    return grasp_pose
```

Listing 2: The code implementation of manipulation concept "Ring_Handle_Lift"

# B  USE OF LARGE LANGUAGE MODELS

In the preparation of this manuscript, large language models (LLMs) were employed as a general-purpose assistance tool. Specifically, we used Gemini and Grok to check for grammar and spelling errors, as well as to suggest refinements in language phrasing and sentence structure for clarity and

conciseness. These suggestions were reviewed, edited, and selectively incorporated by the authors to ensure alignment with the intended meaning and scientific accuracy.

The LLMs did not contribute to the research ideation, methodology design, data analysis, or generation of core content. All intellectual contributions, including hypotheses, experiments, and conclusions, were solely developed by the human authors. We take full responsibility for the final content of the paper, including any potential errors or inaccuracies.

