# OpenReview forum: "Manipulation Concept: Towards Deriving Generalizable and Physics-informed Manipulation Knowledge of Articulated Objects"
_ICLR.cc/2026/Conference — Submitted to ICLR 2026_

### Official Review · Reviewer_rSyi · 2025-10-19

**Soundness:** 3
**Presentation:** 2
**Contribution:** 2
**Rating:** 4
**Confidence:** 4

**Summary:**

This paper introduces Manipulation Concept, an analytic representation that encodes gripper-part interaction skills as parameterized program templates, combining geometric and affordance parameters. A Manipulation Concept Library (MCL) is constructed and integrated with perception modules (VLM, Grounded-SAM) and parameter estimation networks to ground these templates into executable actions. The method is evaluated on the PartNet-Mobility dataset and several real-world objects, showing improved success rates over baselines (Where2Act, ManipLLM, A3VLM) in articulated object manipulation tasks.

**Strengths:**

- Technical quality: The system integrates strong perception and grounding modules into an end-to-end pipeline. Ablations and comparisons support the claimed performance improvements.
- Evaluation: Results on both simulation and real-world settings demonstrate feasibility and robustness.

**Weaknesses:**

- Clarity and writing: The paper introduces many self-defined terms with limited formalization, making it difficult to follow the core idea. The exposition should be more concise and structured.
- System complexity: The pipeline relies on multiple perception and estimation modules, which increases fragility and makes it harder to isolate the contribution of the proposed concept itself.
- Task coverage: Most tasks are relatively simple (e.g., opening covers, lifting handles). The method does not address dynamic or non-prehensile interactions and is limited to quasi-static manipulation.
- Evaluation scope: Real-world results are encouraging but cover a narrow set of scenarios, raising questions about generalization to more complex articulated objects and force-sensitive tasks.

**Questions:**

- Please clarify the formal definition of a manipulation concept and how it differs from prior structured representations (e.g., articulation keypoints, action primitives).
- How would the method extend to dynamic or non-prehensile tasks (e.g., pushing, sliding, compliant doors)?
- Can you provide concrete code/template examples to better illustrate the representation?

---

> ### Author Response · Authors · 2025-11-26
> **Response to Reviewer rSyi**
>
> We sincerely thank the reviewer for the feedback. We are delighted to see that our technical quality, feasibility ,and robustness are recognized. We are fully committed to incorporating the reviewer's feedback, dedicating substantial time to both conducting new experimental analysis and revising the presentation. We respectfully ask for understanding for the slight delay in our response. Below, we provide detailed responses to the concerns and questions.
>
> ---
>
> > **W1: Clarity and writing.**
>
> We thank the reviewer for the comment. We have carefully revised the paper to clarify the formalization, improve conciseness and structure, and thereby make the core idea easier to follow. Specifically, we revise the third paragraph of the Introduction and the first paragraph of Section 3.1 (Manipulation Concept) for better presentation.
>
> **Here we briefly clarify our core idea:**
> This paper's core idea is the introduction of Manipulation Concept, an analytic representation that encodes gripper-part interaction skill as a parameterized program template considering both the object's geometric structure and the gripper's physical characteristics, thereby enabling the derivation of physics-informed actions for articulated object manipulation.
>
> ---
>
> > **W2 : System robustness & Isolation of manipulation concept contribution.**
>
> We sincerely thank the reviewer for this concern.
>
> **Fragility:**
> We respectfully argue that our system design will not increase fragility, for the following two reasons:
>
>
> **1)** In Sec 4.1.3 we provide a system error breakdown to evaluate the impact of each module on the overall task performance. Here we provide a more detailed and fine-grained evaluation of **the accuracy of each module in our pipeline**, where each module receives ground-truth inputs. We compare this with a baseline in which each module receives the predicted output from the preceding module (i.e., without any ground-truth inputs). This setting leads to error accumulation across stages. The results show that the performance gap between the two settings is relatively small, indicating that errors do not accumulate severely across the pipeline.
>
> | Module                     | Metric                        | w/o GT input             | GT input               |
> |----------------------------|-------------------------------|------------------------------|------------------------------|
> | Concept Selection         | Accuracy ↑                     | 90%   |90%   |
> | Part Grounding (Grounding-DINO) | IoU ↑                       | 0.72  |0.78          |         |
> | Part Segmentation (SAM) | IoU ↑                        | 0.86        |0.92         |
> | Structure Parameter Est.   | Relative Size Error ↓               | 15%|11%|
> | Pose Parameter Est.        | ADD-S ↓                        | 0.069 | 0.058 |
>
>
>
> **2)** Although the error accumulation in the pipeline is inevitable, our method allows certain tolerance to possible errors. Here we take *Concept Selection Error* during a manipulation task to "open the cover of a trash can" as an example to describe the tolerance for error occurring in object identification and parameter estimation. When the geometric primitive type is misclassified(*e.g.*, selecting a Cuboid when the true geometry is a Cylinder), our parameter estimation module first predicts geometric parameters to closely approximate the actual shape. Furthermore, successful manipulation can be achieved through the estimation of affordance parameters. The key is that the Manipulation Concept, by providing a prior about part structure and affordance, allows the learning mechanism to participate and adaptively instantiate the concept based on the visual perception, thereby mitigating the impact of geometric variations.
>
>
> **Contribution of Manipulation Concept:**
> We respectfully remind that manipulation concept is the core of our method, whose representation exists throughout our framework, as indicated in Sec 3.2, hence cannot be simply removed. However, its contribution can still be conveniently assessed where we replace the concept with a different representation of actionable part: GAPart [1], which represents the part using its 6D pose and semantic attributes. By keeping all other settings identical for a fair comparison, the results in the table below demonstrate the critical contribution of our proposed concepts.
>
>
> | Representation            | Train success rate (%) | Test success rate (%) |
> |-------------------------------|------------|------------|
> | GAPart [1]      | 41% | 39% |
> | Manipulation Concept      | 59% | 58% |

---

> ### Author Response · Authors · 2025-11-26
> **Response to Reviewer rSyi (continued #1)**
>
> > **W3 & Q2 : Extension to complicated dynamic and non-prehensile tasks.**
>
>
> We sincerely appreciate the reviewer's concern. We have added more real-world manipulation tasks, including *opening the door of a microwave*, *rotating a bottle cap*, *(non-prehensile) pressing a button*, *(non-prehensile) pushing a block*, *(non-prehensile) pushing a drawer*. So far, current experiments have covered diverse tasks (e.g., pressing, pushing, and rotating). This also demonstrates that our method can also be applied to a wide range of non-prehensile manipulation tasks. **Please refer to the folder *additional_video_for_rebuttal* in the supplementary material for more real-world experiment videos.**
>
> For tasks requiring dynamic interaction, we respectfully remind that our method focuses on articulated object manipulation, while dynamic interaction is another specific research field. However, we can extend more physical information such as interaction force and velocity and leverage the pose tracking method (*i.e.*, FoundationPose) to extend the applicability of manipulation concepts to dynamic interaction. We have included this in the future work in our revised version in Appendix A.5.6.
>
> ---
>
> > **W4 : Generalization to more complex articulated objects and force-sensitive tasks.**
>
> We have added more real-world manipulation tasks involving new categories of articulated objects. Notably, the structural complexity of these objects in our real-world evaluation is comparable to that of the objects in PartNet-Mobility [2], which serves as the dataset adopted by representative articulated object manipulation methods [1,3-5]. **Please refer to the folder *additional_video_for_rebuttal* in the supplementary material for real-world experiment videos.**
>
>
> Our approach does not currently aim to address force-sensitive tasks, as our method is vision-based and focuses on object structure and affordance rather than interaction force, like other vision-based articulated object manipulation methods[1,3-5]. Therefore we should not be blamed for the absence of force-sensitive tasks. However, we can add more physical information such as interaction force to extend the applicability of manipulation concepts to force-sensitive tasks. We have added this in our future work in Appendix A.5.6.
>
>
> ---
>
> > **Q1 : Formal definition and novelty of manipulation concept.**
>
> **Formal definition of a manipulation concept:**
>
> The manipulation concept is a novel, structured representation that describes a manipulation skill in the form of a parameterized program template, where the skill refers to the interaction between a part possessing both semantic and geometric properties and a gripper with specific characteristics (e.g., lifting a ring-shaped handle).
>
> Manipulation concept takes two types of parameters as input:
> 1. Geometry parameters: describe the geometric structure of an actionable part.
> 2. Affordance Parameters: describe the gripper-part interaction pattern.
>
> Leveraging these estimated parameterized inputs, the manipulation concept analytically derives the gripper-specific action for robot execution. This computation is achieved by utilizing spatial transformations and part geometric information to compute the grasp pose and force direction of the gripper.
>
>
> **How it differs from prior structured representations:**
>
> Manipulation concept differs from other structured representations in three aspects:
>
>
> 1.  **Part Representation**:
>     - Manipulation concept utilizes geometric primitives to represent actionable parts, considering their **geometric shapes** to provide more actionable information.
>     - Prior part-level representations (*e.g.*, A3VLM [4], GAPartNet [5]) typically rely on 3D bounding boxes, or 6D part poses to represent actionable parts.
> 2. **Explicit Gripper-Awareness:**
>     -  Manipulation concept jointly considers the actionable part and the gripper for interaction. By explicitly modeling the characteristics of gripper, the manipulation concept can derive actions that **physically adhere to the gripper constraints**.
>     -    Prior representations derive action policy using only object-centric information, **without considering the gripper's characteristics.**
> 3.  **Learning Mechanism in Interaction Strategy**:
>     - Manipulation concept's design enables a **learning-based** approach to predict affordance parameters that directly control the gripper’s interaction pattern, enabling flexible and adaptive manipulation.
>     - Prior representations typically rely on **heuristic** interaction strategies for manipulation.
>
> Together, these designs enable more precise, physically feasible, and generalizable manipulation.  We have added the discussion in our revised version in Appendix A.5.2.
>
> ---
>
> > **Q3 : Code/Template Examples.**
>
> We thank the reviewer for raising this critical question. We have provided a concrete code example in the Appendix A.6.

---

> > ### Author Response · Authors · 2025-11-26
> > **Response to Reviewer rSyi (continued #2)**
> >
> > **References**
> >
> > [1] Geng, H., Xu, H., Zhao, C., Xu, C., Yi, L., Huang, S., & Wang, H. (2023). Gapartnet: Cross-category domain-generalizable object perception and manipulation via generalizable and actionable parts. In Proceedings of the IEEE/CVF Conference on Computer Vision and Pattern Recognition (pp. 7081-7091).
> >
> > [2] Xiang, F., Qin, Y., Mo, K., Xia, Y., Zhu, H., Liu, F., ... & Su, H. (2020). Sapien: A simulated part-based interactive environment. In Proceedings of the IEEE/CVF conference on computer vision and pattern recognition (pp. 11097-11107).
> >
> > [3] Mo, K., Guibas, L. J., Mukadam, M., Gupta, A., & Tulsiani, S. (2021). Where2act: From pixels to actions for articulated 3d objects. In Proceedings of the IEEE/CVF International Conference on Computer Vision (pp. 6813-6823).
> >
> > [4] Li, X., Zhang, M., Geng, Y., Geng, H., Long, Y., Shen, Y., ... & Dong, H. (2024). Manipllm: Embodied multimodal large language model for object-centric robotic manipulation. In Proceedings of the IEEE/CVF Conference on Computer Vision and Pattern Recognition (pp. 18061-18070).
> >
> > [5] Huang, S., Chang, H., Liu, Y., Zhu, Y., Dong, H., Gao, P., ... & Li, H. (2024). A3vlm: Actionable articulation-aware vision language model. arXiv preprint arXiv:2406.07549.

---

### Official Review · Reviewer_rMeD · 2025-10-30

**Soundness:** 3
**Presentation:** 3
**Contribution:** 3
**Rating:** 6
**Confidence:** 5

**Summary:**

The paper proposes Manipulation Concept, a representation for articulated-object manipulation. Instead of only predicting “where to act” or only estimating articulation, the method encodes how a parallel gripper should interact with a specific actionable part as a parameterized code template (e.g., Ring_Handle_Lift, Cuboid_Cover_Open). Each concept is defined over (i) geometric parameters (a small set of primitives: cuboid, ring, cylinder, sphere) and (ii) affordance parameters (in (−1, 1), controlling concrete grasp pose / force direction). On top of this representation, the authors build a pipeline: a VLM picks the object / part / action → Grounded-SAM extracts the part point cloud → a point-transformer-style module regresses geometric and affordance parameters → the instantiated concept analytically produces gripper actions. Experiments on PartNet-Mobility (sim) and several real-world articulated objects show clearly higher success rates than Where2Act, ManipLLM, and a re-implemented A3VLM, especially on unseen categories. A useful system-level error breakdown shows that concept selection and part grounding are current bottlenecks.

**Strengths:**

* The authors are going in the right direction by making affordance explicitly gripper-conditioned. A lot of prior “pixel-to-action” or “part-to-action” works conflate “this part is movable” with “this is how my specific end-effector should move it.” This paper makes that mapping explicit: (actionable part, gripper action) → an executable template. For robotics this is the level that’s actually useful.
* Although many submodules are off-the-shelf (Grounded-SAM, FoundationPose, Point Transformer, GMM), this is acceptable in robotics: integration is 50% of the contribution. The paper shows a working stack from language → vision → part → parameters → executable gripper command, in sim and on a real arm. That’s valuable.
* On both train and especially test categories, the method beats Where2Act / ManipLLM / A3VLM under the same parallel-gripper setup (Table 1).
* The progressive “give GT to the next stage” analysis is very informative: it cleanly shows that perception-side errors (concept selection, part grounding) dominate, and that once the concept is right and geometry is right, the analytic template really works. I like this style of system-error decomposition.

**Weaknesses:**

* The whole pipeline assumes that target parts can be simplified to cuboid / ring / cylinder / sphere. That’s elegant, but in real homes/industry we see non-convex, multi-material, undercut, and visually weird handles. Those are precisely the OOD cases where you need affordance generalization, and here the abstraction is at odds with the goal.
* Figure 3 (right) has three “?”: It’s not clear what those three question marks represent.
* You call Table 2 “ablation,” but it’s really cumulative system error decomposition (“what if this stage were perfect?”).
* In autonomous driving there are works that also take the stance “keep a small program / template then fill it,” e.g. “Editable Scene Simulation for Autonomous Driving via Collaborative LLM-Agents,” CVPR 2024 and “Chameleon: Fast–Slow Neuro-Symbolic Lane Topology Extraction,” ICRA 2025. Citing them would make the story stronger.

**Questions:**

The authors say they generate a canonical mesh at a standard pose using the estimated geometric parameters and then send it to FoundationPose. Is this mesh literally just the primitive (e.g., a cylinder with radius R, height H)? If so, how robust is FP when the real part has extra geometry (ribs, a spout, fillets)? How about trying a learned single-view CAD recovery method (e.g., One View, Many Worlds: Single-Image to 3D Object Meets Generative Domain Randomization for One-Shot 6D Pose Estimation) to produce a more realistic surrogate?

---

> ### Author Response · Authors · 2025-11-26
> **Response to Reviewer rMeD**
>
> We thank the reviewer for the detailed and insightful review. We greatly appreciate the recognition of our contribution, the superior experimental performance, and the good presentation and soundness. We have conducted additional experiments and extensively revised our paper, attempting to address the reviewers' questions and suggestions. The effort has resulted in a slight delay in our response, and we kindly ask for the reviewer's understanding. Below, we provide detailed responses to the concerns and questions.
>
>
> ---
>
> > **W1 : Affordance generalization to non-convex, multi-material, undercut, and visually weird target parts.**
>
> Thank you for raising the concern. We humbly clarify that the affordance generalization is available to complex geometries (*i.e.* non-convex, multi-material, undercut, visually weird ones) even when they are not perfectly represented by our primitives.
>
>
> The integration of a structured prior with a learning-based mechanism enables effective handling of parts with non-standard geometries. Specifically, manipulation concept serves as a structured prior that provides physics-informed knowledge to guide the robot action. By explicitly modeling the part’s geometry and affordances through learnable parameters, the manipulation concept effectively defines the target part's feasible interaction space, as mentioned in Sec 3.1. We then employ a learning-based module to directly estimate parameters from point cloud input. This estimation process enables our method to effectively handle local geometric variations and can generalize well to irregular geometries, ensuring successful execution. We have verified this affordance generalizability through experiments on PartNet-Mobility, a dataset of diverse articulated objects containing non-convex and visually complex parts.
>
> ---
>
> > **W2 \& W3 : Presentation issues.**
>
> Thank you for raising the points. "?" refers to "Figure 3", and the "ablation" refers to a "cumulative system error decomposition". We have carefully revised the presentation for better understanding in the updated version.
>
> ---
>
> > **W4 : Additional related works.**
>
> Thanks for the suggestion, and we have cited the two papers in Sec.2.2 to incorporate the ideas of "keeping a small template then fill it", making our story stronger.
>
> ---
>
> > **Q1 : Robustness of FP with primitive input \& Suggestion of using learned CAD recovery method to produce FP input.**
>
> We greatly appreciate the reviewer for raising this insightful question regarding the robustness of FoundationPose (FP) when dealing with complex geometries.
>
> First, the mesh is the primitive as the reviewer assumes.
>
> We conducted a case study to evaluate the performance of FP when the actual part exhibits extra geometry. This analysis is placed in the Appendix A.5.5 (Figure 7 and related text). We found that even when the part contains extra geometry, FoundationPose is still able to predict the pose of the geometric primitive with sufficient accuracy to achieve a reasonable alignment with the actual point cloud. Across the majority of our experiments, the impact of pose error on the manipulation success rate remains within an acceptable tolerance.
>
>
> As suggested by the reviewer, we compare the performance of FP using two different input mesh sources: 1) our geometric primitive, 2) mesh generated by the learned CAD recovery technique "One View, Many Worlds"[1]. Since performance achieved by both approaches is similar and the geometry primitive is required by multiple downstream modules (*i.e.* FP, deriving grasp pose and force direction), it is more self-contained and efficient to use geometry primitive than incorporating a separate CAD recovery method.
>
>
> | method | ADD-S ↓|
> | --- | --- |
> |CAD recovery| 0.055|
> |Ours| 0.058 |
>
> We have included the discussion in our revised version in Appendix A.5.5.
>
> ---
>
> **Reference**
>
> [1] Geng, Z., Wang, N., Xu, S., Ye, C., Li, B., Chen, Z., ... & Zhao, H. (2025). One View, Many Worlds: Single-Image to 3D Object Meets Generative Domain Randomization for One-Shot 6D Pose Estimation. arXiv preprint arXiv:2509.07978.

---

> > ### Comment · Reviewer_rMeD · 2025-11-27
> > **feedback**
> >
> > Thanks for the clarifications and additional experiments. (1) Acceptable. The final action prediction is directly performed on the observed point cloud. (2) I realized my earlier comment had a typo: the three question marks are in Figure 2 (right), not Figure 3; I checked the new version and it still seems unchanged. (3) The FoundationPose case study is interesting; it may well be the case that FP can handle pose estimation reliably even from a clean primitive input.

---

> > > ### Author Response · Authors · 2025-11-27
> > >
> > > Dear Reviewer rMeD,
> > >
> > > Thank you for taking the time to read our rebuttal. We are encouraged to hear your positive feedback.
> > >
> > > We suggest that *the question mark typo* occurs from font missing error. We have further carefully reviewed and replaced all suspicious fonts in figures with commonly supported fonts to ensure correct rendering.
> > >
> > > If there remains any questions, please feel free to share them, and we would be glad to provide responses promptly.
> > >
> > > Best regards,
> > >
> > > Authors of Submission 1343

---

### Official Review · Reviewer_tECJ · 2025-10-31

**Soundness:** 3
**Presentation:** 2
**Contribution:** 1
**Rating:** 4
**Confidence:** 4

**Summary:**

This paper introduces a novel pipeline for general articulated object manipulation with robot grippers, particularly parallel grippers. To address the complexity of articulated objects and gripper interactions, the authors introduce the concept of "manipulation," a library of templates for different object part connections and shapes. To enable generalizable perception, the authors utilize a Vision-Language Model (VLM) to identify actionable parts and the corresponding manipulation concepts. Geometry and affordance parameters are then estimated using a learned neural network. These parameters facilitate precise motion planning, thereby improving both accuracy and success rates. Experiments are conducted in both simulation and real-world settings. The reported success rates demonstrate the superiority of the proposed method over previous approaches.

**Strengths:**

- The authors abstract interactions with articulated objects into a series of manipulation concepts, each described by geometric and affordance parameters. This provides a generalizable and accurate manipulation paradigm.
- The affordance parameters offer greater control over gripper motion, enabling more dexterous and multi-modal manipulation.
- The proposed method is supported by extensive experimentation. It is evaluated on the Where2Act simulation benchmark and a real-world robot arm, showing significant improvements over three baseline methods.

**Weaknesses:**

- The most relevant prior work to this paper is SAGE[1], which also incorporates a VLM and Grounded-SAM to determine actionable part categories and uses learned networks to measure object sizes and poses. While SAGE is mentioned in the related works section, the authors do not clarify the novelty of their approach or provide a comparison of performance in the experiments.
- The manipulation concepts still rely on hand-crafted templates, which can be labor-intensive when scaling to additional object categories.
[1] Haoran Geng, Songlin Wei, Congyue Deng, Bokui Shen, He Wang, and Leonidas Guibas. Sage: Bridging semantic and actionable parts for generalizable manipulation of articulated objects. In ICLR 2024 Workshop on Large Language Model (LLM) Agents.

**Questions:**

- In the geometry parameter estimation, the combination of Point Transformer and MLP seems excessive for the problem. Why not consider traditional parameterization techniques, such as RANSAC?
- Although the paper claims to account for the unique characteristics of different grippers, it focuses only on parallel and suction grippers. The actions of both can be modeled with simple 6-DoF poses. Would the proposed method still apply if extended to fingered or flexible grippers?

---

> ### Author Response · Authors · 2025-11-26
> **Response to Reviewer tECJ**
>
> We sincerely thank for the reviewer's feedback. We are delighted by the recognition of our generalizable and accurate paradigm, the greater control over gripper motion enabling more dexterous and multi-modal manipulation, and the extensive experiments. To enhance our paper and fully incorporate the valuable feedback, we dedicated time to running additional experiments and making extensive revisions. While this process led to a slight delay in our response, we kindly ask for the reviewer's understanding. Below, we provide detailed responses to the concerns and questions.
>
> ---
> >  **W1 :  Novelty justification and performance comparison against SAGE[1].**
>
> Thanks for raising this point. The novelty of our work compared with SAGE lies in three aspects:
>
> 1. **Part Structural Representation:**
>     - SAGE represents an actionable part using its **6D pose**.
>     - As mentioned in Sec.3.1, our manipulation concept represents an actionable part with a parameterized geometric primitive, encoding **more physical information**, including 6D pose, spatial structure, affordance knowledge, enabling more precise manipulation.
> 2. **Explicit Gripper-Awareness:**
>     -    SAGE derives action policies based solely on part's 6D pose, **without explicitly considering grippers' physical constraints**.
>     -    Our manipulation concept considers both sides involved in interaction: the actionable part and the gripper. By explicitly modeling the characteristics of gripper, the manipulation concept can derive grasp poses and force directions that **physically adhere to the gripper constraints**.
> 3. **Learning mechanism in deriving actions:**
>     -    SAGE adopts a **heuristic** strategy to derive action policies.
>     -    As mentioned in Sec 3.2.3, our manipulation concept incorporates a **learning mechanism** to derive action policies, allowing adaptation to more diverse object structures and more robust manipulation.
>
> We further provide the following table to clearly highlight the key differences between our method and SAGE. We have added the discussion about the novelty of our method in Appendix A.5.2.
>
> | Method  | Part Representation | Gripper-Awareness | Interaction Strategy |
> |---|---|---|---|
> | SAGE[1]    | 6D pose  | ✗ | Heuristic Policy |
> | Ours      | Geometric Primitive (including 6D pose, structure, affordance) | ✓ | Learning Mechanism |
>
> A direct comparison with SAGE is not applicable since its code is not open-source. Instead, since SAGE adopts its representation and heuristics from GAPartNet[2], we compare with GAPartNet to evaluate the effectiveness of our key innovations. As shown in the table below, these three aspects contribute to the superior performance of our approach.
>
> | Method | train acc. | test acc. |
> |---|---|---|
> | GAPartNet    | 36%  | 30% |
> | Ours  | 59%  | 58% |
>
> >  **W2 :  Scalability of manipulation concept to additional object categories.**
>
> We respectfully clarify that scaling manipulation concepts requires minimal human labor due to three factors.
>
> 1. **Broad Concept Coverage Minimizes the Need for Crafting New Concepts.** Diverse real-world objects often share geometrically and functionally similar parts, therefore a single concept can offer broad coverage across categories (*e.g.* "Ring Handle" on *bucket*, *kettle*, *pot*, *etc.*). Hence, extending to a new object category requires creating only a few (sometimes even zero) additional concepts. We will open-source the existing manipulation concepts to further reduce the cost to craft new concepts.
> 2. **Crafting New Concepts requires Minimal Labor** because of (1) **Code reusability:** Hand-crafting concepts is efficient in practice, as new concepts can be conveniently constructed by inheriting from and composing reusable functions and classes. For example, a function for *''grasp a ring shape''* can be reused across various concepts, as illustrated by the code example in Appendix A.6. (2) **Visualization tool:** We have developed a visualization tool that provides an interactive display of the manipulation knowledge for each concept, enhancing the efficiency of the manual creation process, as shown in Figure 5.
> 3. **Crafting New Concepts is relatively Scalable.** It is inevitable for articulated object manipulation methods[1-5] to collect training data for optimal performance on additional categories. As discussed above, crafting new concepts requires minimal labor, typically far less than the time for data collection. Therefore, compared with truly labor-intensive data collection, the labor of crafting new concepts is negligible and poses no obstacle to scaling to additional categories.
>
> To further evaluate the scalability of crafting concepts, we recruit 3 college students to craft 4 templates for 4 novel categories (scissors, lighter, pliers, mouse). An average of **2.1 hours** are taken, demonstrating the scalability of concept crafting. **We have added the analysis regarding scalability of crafting concepts to Appendix A.5.4.**

---

> ### Author Response · Authors · 2025-11-26
> **Response to Reviewer tECJ (continued)**
>
> > **Q1 : Our method vs. Traditional parameterization techniques for geometry parameter estimation.**
>
> We thank the reviewer for this question. We have considered traditional parameterization techniques and encountered two obstacles:
> 1. Partial point cloud, the input of parameter estimation, lacks sufficient and global patterns for traditional techniques to establish precise matching, resulting in failure.
> 2. Traditional techniques are more sensitive to occlusion and input noise, which degrades overall performance, especially in real-world experiments.
>
>
> We further compare our learning-based method for parameter estimation against a traditional technique that uses the RANSAC algorithm. Relative Size Error (RSE) is used as metric. Results in the table below demonstrate our superior performance.
>
> | Method | RSE ↓ |
> | --- | ---  |
> | Traditional (RANSAC) | 23% |
> | Ours (Point Transformer + MLP) | **11%**  |
>
> ---
>
> > **Q2 : Applicability to fingered or flexible grippers.**
>
> We thank the reviewer for highlighting this critical aspect.
>
> **The extension of manipulation concept to multi-fingered or flexible hands is feasible**, since the end-effector's specific characteristics can be explicitly encoded in the manipulation concept's definition for gripper-part interaction.
>
>
> Here we provide an example showing how to implement manipulation concept with a 5-fingered dexterous hand to complete "pulling the ring handle with full hand" task in five steps. **Please refer to Appendix A.5.1 for detailed analysis and visualization.**
>
>
> 1. **Initial Hand Pose:** The hand's root pose is initialized such that the palm faces the ring, with the palm normal aligned parallel to the ring's central axis, and the fingertips aligned parallel to the ring's axial direction.
> 2. **Fingers Initial Configuration:** For the four fingers (excluding the thumb), the adduction/abduction joint (sideways movement) is set to 0, ensuring the fingers are laterally aligned. Based on the ring's axial thickness and radial thickness, a collective bending angle is calculated and proportionally distributed across the joints to conform the four fingers to the ring's surface. The thumb's metacarpophalangeal joint (the second joint from the fingertip) is set to 0.8, allowing the thumb to be parallel to the other four fingers and make contact with the ring's surface.
> 3. **Affordance Parameters:** We design seven affordance parameters.
>     (a), (b), \(c): Control the angular adjustments in the four fingers (excluding the thumb), specifically, the adjustments in the spread angle of adduction/abduction joint (sideways movement), the bending angle of the distal (fingertip) and middle joints (the two joints near the fingertip), and the bending angles of the proximal joints (closest to the palm). (d): Controls the rotation of the thumb along the ring's surface. (e), (f), (g): Control the palm's relative translation and rotation with respect to the ring (consistent with the parallel gripper, as shown in Figure 2 of the main paper).
> 4. **Force direction:** The required force direction for the dexterous hand aligns with that of the parallel gripper.
>
> With such configuration, a dexterous hand can grasp a ring-shaped handle and pull it.
>
> ---
>
> **References**
>
> [1] Haoran Geng, Songlin Wei, Congyue Deng, Bokui Shen, He Wang, and Leonidas Guibas. Sage: Bridging semantic and actionable parts for generalizable manipulation of articulated objects. In ICLR 2024 Workshop on Large Language Model (LLM) Agents.
>
> [2] Geng, H., Xu, H., Zhao, C., Xu, C., Yi, L., Huang, S., & Wang, H. (2023). Gapartnet: Cross-category domain-generalizable object perception and manipulation via generalizable and actionable parts. In Proceedings of the IEEE/CVF Conference on Computer Vision and Pattern Recognition (pp. 7081-7091).
>
> [3] Mo, K., Guibas, L. J., Mukadam, M., Gupta, A., & Tulsiani, S. (2021). Where2act: From pixels to actions for articulated 3d objects. In Proceedings of the IEEE/CVF International Conference on Computer Vision (pp. 6813-6823).
>
> [4] Li, X., Zhang, M., Geng, Y., Geng, H., Long, Y., Shen, Y., … & Dong, H. (2024). Manipllm: Embodied multimodal large language model for object-centric robotic manipulation. In Proceedings of the IEEE/CVF Conference on Computer Vision and Pattern Recognition (pp. 18061-18070).
>
> [5] Huang, S., Chang, H., Liu, Y., Zhu, Y., Dong, H., Gao, P., … & Li, H. (2024). A3vlm: Actionable articulation-aware vision language model. arXiv preprint arXiv:2406.07549.

---

### Official Review · Reviewer_v61i · 2025-10-31

**Soundness:** 3
**Presentation:** 3
**Contribution:** 3
**Rating:** 6
**Confidence:** 5

**Summary:**

The manuscript introduces a novel analytic representation called Manipulation Concept, which encodes gripper-based manipulation skills as parameterized program templates. Each concept formalizes the interaction between an actionable part (e.g., cuboid door, ring handle) and a gripper action (e.g., push, lift), linking geometric and semantic information with executable robot actions.

The authors build a Manipulation Concept Library (MCL) and propose an end-to-end framework that integrates:

1/ A vision-language model (VLM) for concept selection based on language instructions and visual inputs;
2/ A Grounded-SAM module for actionable part grounding;
3/ A parameter estimation network predicting geometric and affordance parameters;
4/ A programmatic instantiation step that outputs precise gripper poses and force directions for execution.

Experiments on the PartNet-Mobility dataset and real-world articulated objects demonstrate that the approach outperforms prior state-of-the-art methods (e.g., Where2Act, ManipLLM, A3VLM) in success rate and generalization ability.

**Strengths:**

1/ The “Manipulation Concept” framework provides a clean, modular, and interpretable way to bridge geometric reasoning, affordances, and physical execution — something often lacking in purely data-driven manipulation approaches.
2/ The method generalizes across unseen object categories, with substantial improvements in both simulated and real-world settings, indicating that the analytic abstraction is meaningful and transferable.
3/ The paper is well written and logically organized, with clear diagrams (e.g., Fig. 1–3) illustrating the concept structure and pipeline.

**Weaknesses:**

1/ The approach focuses mainly on parallel grippers. Although the authors briefly mention possible extensions to suction grippers, broader applicability to multi-fingered hands or tool use remains unexplored.
2/ While programmatic templates improve interpretability, they require manual definition and careful parameterization. It’s unclear how scalable the approach is when facing highly deformable or irregular geometries not well approximated by simple primitives.
3/ The multi-stage process (VLM → segmentation → pose estimation → parameter inference) could introduce latency. Real-time performance metrics are not discussed.

**Questions:**

Please see the weakness section.

---

> ### Author Response · Authors · 2025-11-26
> **Response to Reviewer v61i**
>
> We thank the reviewer for the constructive and insightful feedback. We greatly appreciate the recognition of our clean and interpretable framework, the generalization ability, and the clear presentation. To better address the reviewer's questions and further improve our paper according to the reviewer's suggestions, we conducted many additional experiments and carefully revised our paper, which results in a slight delay in our response. We kindly ask for the reviewer's understanding. Below, we provide detailed responses to the concerns and questions.
>
> ---
> > **W1 : Applicability to multi-fingered hands or tool use.**
>
> We thank the reviewer for raising this critical point.
>
> Our method is extensible to multi-fingered hands and tool use. Since the manipulation concept can explicitly encode the end-effector's specific characteristics, multi-fingered hands are applicable. Tool use is also supported, as the tool can be modeled as an extension of the end-effector.
>
>
> Here we provide an example showing how to implement a manipulation concept with a 5-fingered dexterous hand to complete "pulling the ring handle with full hand" task in four steps. **Please refer to Appendix A.5.1 for detailed analysis and visualization.**
>
> 1. **Initial Hand Pose:** The hand's root pose is initialized with the palm facing the ring. Specifically, the palm normal is aligned parallel to the ring's central axis, and the fingertips are aligned parallel to the ring's axial direction, preparing the initial grasp.
> 2. **Fingers Initial Configuration:** Lateral alignment for the four fingers (excluding the thumb) is achieved by zeroing their adduction/abduction joints. The necessary collective bending angle is then calculated based on the ring's axial and radial thickness. This angle is proportionally distributed across the individual joints to ensure the fingers accurately conform to the ring’s surface. The thumb's metacarpophalangeal joint is set to a flexion value of 0.8 radians, allowing the thumb to be parallel with the other four fingers and make firm contact with the ring's surface.
> 3. **Affordance Parameters:** We design seven sets of affordance parameters, categorized by their control function: **(a):** abduction/Adduction angle (sideways movement) of the metacarpophalangeal joints (the second joint from the fingertip) in the four fingers, **(b):** the bending angle of the distal (fingertip) and middle joints (the two joints near the fingertip), **\(c):** the bending angles of the proximal joints, **(d):** the rotation control of the thumb along the ring's surface, **(e)-(g)** palm's relative translation and rotation with respect to the ring (consistent with the parallel gripper).
> 4. **Force direction:** The required force direction for the dexterous hand follows that of the parallel gripper.
>
> With such configuration, a dexterous hand can grasp a ring-shaped handle and pull it.

---

> ### Author Response · Authors · 2025-11-26
> **Response to Reviewer v61i (continued)**
>
> > **W2 : Scalability of programmatic templates to deformable or irregular geometries.**
>
>
> Thanks for the insightful point.
>
> First, we clarify that in this paper we focus on **articulated objects** rather than **deformable ones**, as mentioned in our title and abstract. However, we will explore concept representation for deformable objects and integrate them to our framework, as stated in our future work (Sec A.5.6).
> Second, we elaborate that our approach is **scalable for irregular geometries** for three reasons:
> 1. **Only a few (sometimes even zero) new concepts are needed to construct to cover irregular geometries.** We have created 52 manipulation concepts that cover 18 common types of actionable parts. Existing concepts can cover  diverse geometries, including irregular ones, which is validated by the high success rates of our manipulation tasks on PartNet-Mobility[1], a dataset comprising various articulated objects that encompass a wide range of geometric characteristics including irregular ones.
> 2. **Creating new concepts for irregular geometries is scalable.**  The programmatic nature of our concepts allows for inheritance and composition, facilitating scalable extension through code reuse, as illustrated by the code example in Appendix A.6. Furthermore, we develop a visualization tool to help create manipulation concepts, as shown in Figure 5. Practically, it takes 2.1 hours to create a new concept, which is scalable. Please refer to Appendix A.5.4 for detailed analysis of scalability. We will open-source our concepts to further reduce the effort for the community to create new concepts for irregular geometries.
> 3. **Our framework design can alleviate the gap in representing irregular geometries.** As mentioned in Sec 3.1, the geometry knowledge (possibly not well approximated) indicates the affordance knowledge (grasp pose and force direction), which is then grounded via a learning mechanism that estimates the precise affordance parameters. The learning module uses the actionable part's point cloud (with irregular geometry) as input, enabling our method to inherently accounts for irregular geometry and derive affordance knowledge adaptive to irregular geometry, hence alleviates the gap in representing irregular geometries.
>
> We have included this discussion in our revised version in Appendix A.5.3.
>
> ---
>
> > **W3 : Real-time performance.**
>
> Thanks for raising the point. We humbly clarify that our system is decoupled to a **initialization phase** and an **execution phase**(begins when the robot receives the grasp pose and force direction, ends with the robot initiating motion). The initialization is a one-time effort to perceive the workspace that should not count to execution phase latency, and our execution phase is real-time.
>
> Here we provide a statistic of the time cost of each module in our multi-stage process on a single NVIDIA 3090 GPU. The results are averaged on 10 runs. The execution phase latency for execution stage requires about **0.09** seconds, which does not impede real-time execution. The latency for all stages is about 6.97 seconds, similar to ManipLLM, demonstrating our system's practicality. Detailed analysis is added to Appendix A.2.4.
>
>
> | Phase                | Stage                                | Average Latency (s) |
> |----------------------|--------------------------------------|---------------------|
> | Initialization | Concept Selection (VLM-based)        | 3.19                |
> |                      | Part Grounding & Segmentation        | 1.55                |
> |                      | Geometric Parameter Estimation       | 1.70                |
> |                      | Affordance Parameter Estimation      | 0.44                |
> | Execution      |   Execution Phase Latency                | 0.09                |
> |      Total           | **Total Latency**                    | 6.97                |
> |      Total             | ManipLLM (Baseline)                  | 6.72                |
> |      Total              | A3VLM (Baseline)                     | 60.10               |
>
> ---
>
> **References**
>
> [1] Xiang, F., Qin, Y., Mo, K., Xia, Y., Zhu, H., Liu, F., ... & Su, H. (2020). Sapien: A simulated part-based interactive environment. In Proceedings of the IEEE/CVF conference on computer vision and pattern recognition (pp. 11097-11107).

---

> > ### Comment · Reviewer_v61i · 2025-11-27
> >
> > I appreciate the efforts the authors made during rebuttal. My major concerns are addressed.

---

> > > ### Author Response · Authors · 2025-11-27
> > >
> > > Dear Reviewer v61i,
> > >
> > > Thank you for taking the time to read our rebuttal. We sincerely appreciate your efforts and are glad to address your major concerns. Please feel free to share any additional questions, and we will make sure to follow up.
> > >
> > > Best regards,
> > >
> > > Authors of Submission 1343

---

### Author Response · Authors · 2025-12-03
**Summary of Our Work and Rebuttal**

Dear ACs,

We sincerely thank all reviewers for their constructive feedback and for recognizing **1) Novelty.** Manipulation Concept representation bridges geometric reasoning, gripper-conditioned affordance with physical execution (`v61i`, `rMeD`) and provides a generalizable, accurate manipulation paradigm (`tECJ`);  **2) Effective framework.** Our framework integrates many submodules, demonstrating strong technical quality (`rSyi`) and practical value (`rMeD`); **3) Superior performance** (`v61i`,`tECJ`,`rMeD`,`rSyi`). Generalizability (`v61i`,`rMeD`) and robustness (`rSyi`) of our method;  **4) Good presentation** (`v61i`,`rMeD`).

Reviewers `v61i` and `rMeD` initially give positive ratings (both rating 6, confidence 5). Our rebuttal has addressed the major concerns of both Reviewer `v61i` and  `rMeD`, as reflected in their positive responses. The other two reviewers (`tECJ`,`rSyi`) initially give  (both rating 4, confidence 4), and have not provided responses. Below we summarize our work and the rebuttal phase, including the major concerns and how we address them.

## **Summary of Our Work**
We introduce **Manipulation Concept**, **a novel analytic representation** that **bridges object structural properties and gripper characteristics with task-oriented interactions** for articulated object manipulation. Each concept encodes gripper-part interaction skills into parameterized program templates. We further construct concept library and propose a framework to ground the concepts for **robust and generalizable articulated object manipulation**. Experiments show that our method achieves **superior performance over existing baselines**, and **generalizes well** to unseen categories.

## **Summary of Rebuttal**

> Applicability of Manipulation Concept to more end-effectors or tool use. (`v61i`-W1, `tECJ`-Q2)

Using the "pulling a ring handle" task, we detail the **extension of the Manipulation Concept to a 5-fingered hand**, including the parameters and action design. We also **clarify tool use** as end-effector extension. ***Reviewer `v61i`'s reply suggests that we have addressed this concern.***

> Novelty of Manipulation Concept compared to other representations (e.g., GAPart). (`tECJ`-W1, `rSyi`-Q1)

The Manipulation Concept's novelty is threefold: 1) It **encodes richer physical information** (spatial, affordance, actionable info) beyond 6D poses. 2) It **explicitly models gripper for interaction**, deriving actions that physically adhere to gripper constraints. 3) It **adopts a learning mechanism rather than heuristics**, allowing generalization to diverse object categories. We also provide a **quantitative evaluation**, showing its superiority compared with prior representations.

> Affordance Generalization. (`rMeD`-W1)

The Manipulation Concept's affordance **generalizes well** because our method estimates affordance parameters from the object part's point cloud using a learning mechanism, **handling geometric variations and ensuring robust generalization**. ***Reviewer `rMeD`'s response shows we have addressed the concern.***

> Scalability of Manipulation Concept to new categories (`v61i`-W2, `tECJ`-W2) with irregular geometries (`v61i`-W2).

Our Manipulation Concept is scalable for three reasons: **1) Broad coverage of existing concepts minimizes the need for crafting new concepts.** **2) concept creation requires minimal labor** due to its programmatic nature. **3) The effort to craft concepts is negligible compared to inevitable data collection**. **For irregular geometries, scalability is ensured** by utilizing learning mechanisms on the object part's point cloud, which enables adaptive action generation. ***The response of Reviewer `v61i` suggests our rebuttal has addressed the concern.***

> System robustness \& Manipulation Concept contribution. (`rSyi`-W2)

System robustness is demonstrated via a **detailed module evaluation** (ground-truth *v.s.* accumulative errors) and **a case study**, both showing **minimal performance degradation**. Given the Manipulation Concept is core of our method, we evaluate its contribution through an ablation study (replacing it with GAPart). Its superior performance confirms its critical contribution.

> Experiment on non-prehensile, dynamic, complex and force-sensitive tasks, and complex objects. (`rSyi`-W3,W4,Q2)

We first argue that we focus on **articulated object manipulation**, following the experiment settings of prior research (Where2Act, ManipLLM, A3VLM). **Dynamic, force-sensitive tasks are outside this scope, therefore we should not be blamed for the absence of these experiments.** To mitigate concerns, we add supplementary real-world videos covering more complex and non-prehensile tasks.

---
We are deeply grateful to every AC and reviewer for their dedicated effort and excellent work. The review phase has significantly improved the quality of our work. We hope this summary assists the AC's work.

Best Regards,

Authors of Submission 1343

---

### Author Response · Authors · 2025-12-03
**Summary of Revisions**

We appreciate all the reviewers' effort in providing insightful and constructive feedback. To better clarify the superiority of our method, demonstrate our contribution, and make the paper easier to understand, we have comprehensively revised our paper following the reviewers' suggestions. The updates in the paper are **highlighted in blue**. Here we provide a **summary for the key updates**.

- **Main Paper**
  - **Formalization of Manipulation Concept** in Sec.1 and Sec.3.1, making our core idea easier to follow. (`rSyi`-W1)
  - **Specification of section headings** (Sec.4.1.3 and 4.1.4) for better understanding, and **fixing typos**. (`rMeD`-W2, W3)
  - **Applicability of Manipulation Concept to various types of end-effectors** in Sec 4.1.4. (`v61i`-W1, `tECJ`-Q2)
  - More comprehensive **related works**. (`rMeD`-W4)
  - **Reformatting of Figure.1-4** to more clearly demonstrate our method.

- **Appendix**
  - Sec A.2.3: Additional analysis regarding **inference time**. (`v61i`-W3)
  - Sec A.5.1: **Extensibility** of the manipulation concept (illustrated with a five-fingered dexterous hand). (`v61i`-W1, `tECJ`-Q1)
  - Sec A.5.2: **Clarification of novelty** through comparison with prior structured representations. (`tECJ`-W1, `rSyi`-Q1)
  - Sec A.5.3: Additional discussion on the robustness when dealing with **irregular geometries**. (`v61i`-W2, `rMeD`-W1)
  - Sec A.5.4: Additional discussion on the **scalability** of Manipulation Concept creation. (`v61i`-W2, `tECJ`-W2)
  - Sec A.5.5: Additional robustness analysis of the **pose estimation module**. (`rMeD`-Q2)
  - Sec A.5.6: Expanded discussion of **limitations and future work**. (`v61i`-W1, `tECJ`-Q1, `rSyi`-W3,W4,Q2)
  - Sec A.6: **Code example** of a manipulation concept for better illustration. (`rSyi`-Q3)

- **Supplementary materials:**
  - **Five more video demonstrations** of real-world manipulation tasks involving non-prehensile tasks (`rSyi`-W3,Q2) and complex articulated objects. (`rSyi`-W4)

---

### Meta-Review · Area_Chair_7srk · 2026-01-09

**Summary:**

This paper received 4 mixed reviews (two Borderline Accept and two Borderline Reject).

Most reviewers agreed that the paper is well-written and the experiments are extensive.

On the contrary, the main concerns raised by the reviewers include, but not limited to:
- incremental contributions compared with the prior works
- reliance on hand-crafted templates
- too complex design of the pipeline
- limited scenarios

After the discussion phase, the two reviewers with negative ratings maintain the original score. In sum, I'm inclined to recommend it for a (borderline) rejection.

**Reviewer Scores:**

The two reviewers with the negative initial ratings may maintain their scores.

---

### Decision · Program_Chairs · 2026-01-26

Reject